# Can Textual Gradient Work in Federated Learning?

**Minghui Chen**[1,2,3], **Ruinan Jin**[1,2], **Wenlong Deng**[1,2], **Yuanyuan Chen**[3], **Zhi Huang**[4], **Han Yu**[3], **Xiaoxiao Li**[1,2]
[1]The University of British Columbia    [2]Vector Institute
[3]Nanyang Technological University    [4]University of Pennsylvania

## Abstract

Recent studies highlight the promise of LLM-based prompt optimization, especially with TextGrad (Yuksekgonul et al., 2024), which automates "differentiation" via texts and backpropagates textual feedback provided by LLMs. This approach facilitates training in various real-world applications that do not support numerical gradient propagation or loss calculation. It opens new avenues for optimization in decentralized, resource-constrained environments, suggesting that users of black-box LLMs (*e.g.*, ChatGPT) could enhance components of LLM agentic systems (such as prompt optimization) through collaborative paradigms like federated learning (FL). In this paper, we systematically explore the potential and challenges of incorporating textual gradient into FL. Our contributions are fourfold. **Firstly**, we introduce a novel FL paradigm, Federated Textual Gradient (`FedTextGrad`), that allows FL clients to upload their locally optimized prompts derived from textual gradients, while the FL server aggregates the received prompts through text summarization. Unlike traditional FL frameworks, which are designed for numerical aggregation, `FedTextGrad` is specifically tailored for handling textual data, expanding the applicability of FL to a broader range of problems that lack well-defined numerical loss functions. **Secondly**, building on this design, we conduct extensive experiments to explore the feasibility of federated textual gradients. Our findings highlight the importance of properly tuning key factors (*e.g.*, local steps) in FL training to effectively integrate textual gradients. **Thirdly**, We highlight a major challenge in federated textual gradient aggregation: retaining essential information from distributed prompt updates. Concatenation often produces prompts that exceed the LLM API's context window, while summarization can degrade performance by generating overly condensed or complex text that lacks key context. **Last but not least**, in response to this issue, we improve the vanilla variant of `FedTextGrad` by providing actionable guidance to the LLM when summarizing client prompts by leveraging the Uniform Information Density principle. Such a design reduces the complexity of the aggregated global prompt, thereby better incentivizing the LLM's reasoning ability. Through this principled study, we enable the adoption of textual gradients in FL for optimizing LLMs, identify important issues, and pinpoint future directions, thereby opening up a new research area that warrants further investigation. Our code is available on `https://github.com/ubc-tea/FedTextGrad`.

## 1 Introduction

Large Language Models (LLMs) (Zhao et al., 2023), such as GPT (Brown, 2020), Gemini (Team et al., 2023) and LLaMa (Touvron et al., 2023), have become the foundational backbone of modern natural language processing (NLP) systems. These models often require fine-tuning to enhance their responsiveness to specific tasks. While existing open datasets play an important role in LLM tuning, the vast amount of privately owned, potentially sensitive data continuously generated by end devices represents a significant, yet largely untapped, pool of samples for LLM fine-tuning.

To adapt to this reality, federated learning (FL) (McMahan et al., 2017) offers a promising privacy-preserving framework for collaboratively fine-tuning LLMs with distributed, privately owned data. To address the efficiency demands and black-box nature of many involving LLM APIs (Achiam et al., 2023), recent advancements in zeroth-order optimization (Qin et al., 2023; Fang et al., 2022) are beginning to provide useful tools for exploring this avenue. However, these methods generally rely on numerical loss calculations to estimate gradients Balasubramanian & Ghadimi (2022), which is infeasible when using black-box LLM APIs, where the loss definition is unclear (Yang et al., 2024b) and only textual feedback (e.g., human feedback in ChatGPT (Achiam et al., 2023)) is available (Yuksekgonul et al., 2024).

Recently, LLMs have been demonstrated as effective optimizers (Pryzant et al., 2023; Liu et al., 2024; Yang et al., 2024b), capable of automatically refining prompts step-by-step to enhance performance (Shinn et al., 2024), while providing informative and interpretable natural language criticism to the variables to guide how the variables should update. As a representative method, TextGrad (Yuksekgonul et al., 2024) enables automatic "differentiation" through text, allowing the backpropagation of textual feedback to improve individual components of a compound LLM agentic system without relying on gradients or numerical calculations. While TextGrad offers substantial advantages in traditional centralized machine learning settings, its adaptation to FL environments remains unexplored. In this paper, we seek to answer the exploratory question:

> *Can textual gradient work under federated learning settings?*

Our contributions are fourfold, as outlined below.

**Adapting:** To facilitate **textual gradient** operations in FL environments, we propose a **first-of-its-kind** `FedTextGrad` method. Under this method, each FL client is equipped with TextGrad-based textual gradients during local training. Instead of uploading model parameters like in classical FL (e.g., FedAvg (McMahan et al., 2017)), clients upload their optimized local prompts to the FL server. The server then performs prompt aggregation through concatenating and summarizing clients' local prompts, and redistribute the global prompt back to the clients for further training.

**Investigating:** With `FedTextGrad`, we then conduct experimental studies across various LLMs and configurations to empirically investigate its relative performance under FL settings compared to TextGrad in centralized settings on a range of reasoning tasks. During this process, we study the impact of key factors—such as **local update epochs**, **batch size**, **number of clients**, and **data heterogeneity**—on the performance of our framework.

**Uncovering:** Through our empirical investigation, we have identified a key challenge for federated textual gradient **aggregation**: *preserving critical information in distributed prompt updates*. Concatenation-based prompt aggregation can produce excessively long prompts that exceeds the LLM API's context window, while summarization-based prompt aggregation often degrades performance by generating overly complex and densely packed texts. This is currently the key hurdle hindering the adoption of textual gradient in FL settings.

**Improving:** To address this challenge, we develop an key insight that uneven distribution of information within the summarized prompts is the root cause. We then introduce an enhanced summarization method based on the **Uniform Information Density (UID) principle** to ensure more balanced information distribution across the summarized global prompt. It improves prompt aggregation in `FedTextGrad` by maintaining a uniform information density, resulting in shorter aggregated prompts that preserve critical contents without sacrificing model performance.

**Related work:** A detailed literature review is provided in App. A.
*FL for LLMs.* As LLMs have achieved significant success in centralized learning, there is a growing interest in adapting FL to accommodate the fine-tuning of pre-trained LLMs (Ren et al., 2024), particularly to supplement the publicly available data with privately owned datasets Jin & Li (2023). In response, several frameworks have emerged recently, including OpenFedLLM Ye et al. (2024) and FederatedScope-LLM Kuang et al. (2024). Moreover, advanced methods such as FedbiOT Wu et al. (2024) which safeguards model ownership, and FFA-LoRA Sun et al. (2024) which enhances performance under differential privacy constraints, are being developed to optimize LLM training in federated environments.
*LLMs as Optimziers.* Recent research has turned towards leveraging *LLMs as optimizers* in black-

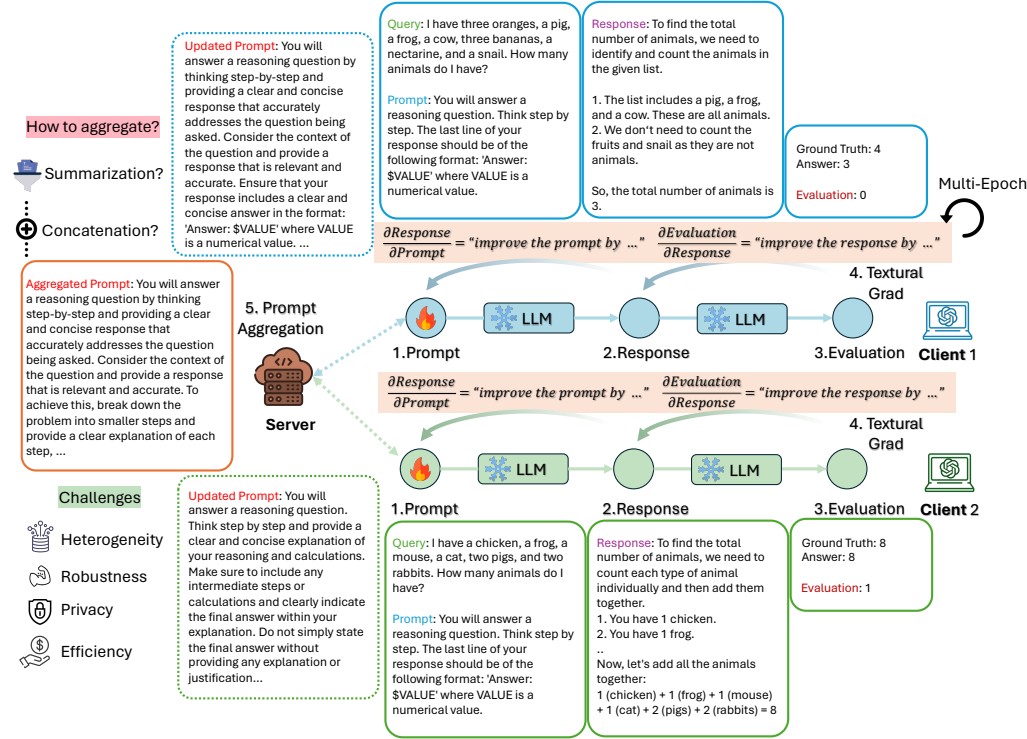

Figure 1: Illustration of `FedTextGrad`, where the upper part (blue boxes) and the lower part (green boxes) represent two different clients. *Within each client*, circles represent the prompts, and boxes represent the LLM. `FedTextGrad` consists of four steps for local updating, proceeding from left to right. In `step-1 (Prompt)`, the client is tasked with answering the *Query* by initializing a *Prompt* to the LLM to obtain a response. Then, in `step-2 (Response)`, the LLM performs multi-step reasoning (e.g., CoT) and generates a Response. In `step-3 (Evaluation)`, the Response is evaluated against the ground truth by the LLM, and a Evaluation score is generated. Finally, in `step-4 (Textual Grad)`, the *Prompt* is updated "backward" based on feedback from the LLM. After this, the client sends the *Updated Prompt* to the server. *On the server-side*, the collected prompts from all clients are aggregated by the server, which acts as a trusted third party, and then sent back to the clients, as shown in `step-5`. Two aggregation strategies are available: simply concatenating the prompts or using the server-side LLM to summarize them. The system iteratively performs local updates (multiple local epochs of `steps 1-4`) and global aggregation (`step-5`) for optimization in the FL system.

box settings (Yang et al., 2023). The foundation of this concept stems from the ability of LLMs to simulate human decision-making. Zheng et al. (2023) benchmarked the behavior of LLMs and human decisions, finding that modern LLMs align closely with human judgment. Building on this, Yang et al. (2024b) proposed *optimization by prompting*, where LLMs generate new solutions based on a prompt that includes previously generated solutions.Ma et al. (2024) further investigated whether LLMs are effective prompt optimizers. Tools like DSPy Khattab et al. (2023) and Pro-TeGi Pryzant et al. (2023) introduced programmatic frameworks for optimizing LLM-based APIs, achieving performance gains across tasks such as question answering and prompt refinement. These new solutions are then assessed and incorporated into the prompt for the next optimization step.

## 2 THE PROPOSED `FedTextGrad` METHOD

In this section, we first provide background on TextGrad, including its forward operation, backpropogation and how LLM-as-the-optimizer can be integrated with TextGrad (Section 2.1). Next, we explain the analogy between textual and numerical gradients, and describe its extension into the FL setting - `FedTextGrad` (Section 2.2). Finally, we present preliminary results across various LLM APIs using TextGrad and `FedTextGrad`, highlighting the performance drops (Section 2.3).

## 2.1 PRELIMINARIES ON TEXTGRAD.

TextGrad is a framework that leverages LLMs for iterative prompt optimization through natural language feedback, combining (1) a forward operation to generate and evaluate responses, (2) backpropagation-like updates using textual gradients, and (3) LLM-based optimization via Textual Gradient Descent to refine prompts effectively across tasks.

**Forward Operation of TextGrad.** As illustrated in Figure 1, the forward operation of TextGrad take input query and the *Prompt* (parameter to be optimized) to a fixed LLM to generating responses. This *Response* is then concatenated with the *Evaluation Instruction* to form the input for the next LLM call for evaluation. The structure of the computational graph can be expressed as:

$$\text{Query} + \text{Prompt} \xrightarrow{\text{LLM}} \text{Response} + \text{Evaluation Instruction} \xrightarrow{\text{LLM}} \text{Evaluation},$$

where $+$ denotes the concatenation operation. The depth of this computational graph can be extended by adding intermediate response nodes before performing the final evaluation step. This extension accommodates a more complex step-by-step reasoning chain, which is analogous to adding more layers in a deep neural network.

**Backpropagation of TextGrad.** Based on the output of the forward operation, TextGrad proceeds with backpropagation to update the *Prompt* by calculating $\partial\text{Evaluation}/\partial\text{Prompt}$ using the Chain Rule – first computing the 'gradient' with respect to the Response, $\partial\text{Evaluation}/\partial\text{Response}$, by collecting feedback on the *Response* from the *Evaluation*; then $\partial\text{Response}/\partial\text{Prompt}$, by obtaining prompt updates from Response using LLMs. The textural gradient represents natural language feedback, such as: "This response can be improved by...", guiding the adjustment of variables (*e.g.*, the *Prompt*) to optimize the downstream objective, similar to how numerical gradients function in traditional optimization. This approach allows for the iterative refinement of the *Prompt*, analogous to the use of numerical gradients in backpropagation to optimize neural network weights.

**LLMs-as-Optimizers in TextGrad.** After obtaining the 'gradient' ($\partial\text{Evaluation}/\partial\text{Prompt}$), Textual Gradient Descent (TGD) leverages LLMs to update the *Prompt*, by iteratively refining it using the obtained textual 'gradient', similar to the backpropagation process in neural networks. The update rule for the *Prompt* is:

$$\text{Prompt}_{\text{new}} = \text{TGD.step}\left(\text{Prompt}, \partial\text{Evaluation}/\partial\text{Prompt}\right), \tag{8}$$

where textual gradients inform the optimization process. Essentially, `TGD.step(·)` is implemented through an LLM call using a predefined instruction template: "Below are the criticisms on {Prompt}: {$\partial\text{Evaluation}/\partial\text{Prompt}$}. Incorporate the criticisms and generate an updated prompt."

## 2.2 FROM TEXTGRAD TO `FedTextGrad`.

We introduce `FedTextGrad`, a novel adaptation of TextGrad for FL environments. While our initial demonstration focuses on prompt optimization (Pryzant et al., 2023), the methodology is versatile and can be applied to a wide range of tasks such as retrieval-augmented generation (Lewis et al., 2020) and tool use (Schick et al., 2024) supporting federated LLM agentic systems. `FedTextGrad` extends TextGrad by integrating textual gradient-based optimization into FL client local training. In this setup, clients optimize their local prompts using LLM-generated textual gradients, sharing these prompts instead of raw gradient updates with the central server. It mirrors *FedAvg* (McMahan et al., 2017), where local model updates are aggregated at the server. Rather than aggregating numerical gradients, `FedTextGrad` aggregates natural language prompts across clients. The key innovation in `FedTextGrad` is enabling collaborative textual optimization in FL settings, where prompts are iteratively improved by individual FL clients and then aggregated to form a global prompt. The challenge here lies in defining an effective aggregation strategy for these local prompts. We first explore intuitive methods such as concatenation and summarization to evaluate their effectiveness across various LLM APIs and FL settings.

## 2.3 `FedTextGrad` FRAMEWORK DESCRIPTION

The `FedTextGrad` framework iteratively refines prompts through (1) client-specific updates using textual gradients, (2) server-side aggregation into a global prompt, and (3) redistribution to clients

across communication rounds. The detailed process, outlined in Algorithm 1, follows these steps:

1. **Client Prompt Updates:** (Algorithm 1, steps 12-18): Each client $i$ receives the global prompt $P^t$ and fine-tunes it using its local dataset $\mathcal{D}_i$. Textual gradients generated by the LLM guide this local optimization, producing an updated prompt $P_i^t$, which captures the unique distribution of each client's data.

2. **Server Prompt Aggregation** (Algorithm 1, step 9): The server collects the updated prompts $P_i^t$ from all clients and aggregates them into a new global prompt $P^{t+1}$. Aggregation strategies such as concatenation or summarization are used to integrate client updates.

3. **Global Prompt Distribution** (Algorithm 1, steps 6-8): The server then distributes the updated global prompt $P^{t+1}$ to all clients. This iterative process continues across several communication rounds, with each iteration refining the global prompt based on client-specific updates.

This iterative framework enables prompt updates at both local and global levels, ensuring the model adapts effectively to heterogeneous client environments.

## 3 EXPERIMENTAL INVESTIGATION

### 3.1 EXPERIMENT SETTINGS

**Data and Tasks.** We evaluate `FedTextGrad` on prompt optimization across three key tasks from the *BBH* benchmark (Srivastava et al., 2022): 1) *BBH Object Counting*, 2) *BBH Multistep Arithmetic*, and 3) *GSM8k Math Problem* (Cobbe et al., 2021). They are well-suited for assessing the effectiveness of prompt optimization in complex reasoning scenarios. For each dataset, we split it into *training*, *validation*, and *test* sets. We adopt the dataset preprocessing methodology outlined in (Yuksekgonul et al., 2024). The training set is used for prompt optimization. The validation set is used for prompt selection and hyper-parameter tuning. The test set is used for reporting the final performance, thereby ensuring fair and rigorous evaluation.

**Model and Setup.** For our experiments, we use the Llama-3.1-8B model (Dubey et al., 2024) for prompt optimization, serving as both the inference engine and the optimizer within our framework. Unless otherwise specified, we use a default batch size of 3 with 3 local steps for tuned hyper-parameters, with batches sampled randomly with replacement. After each iteration,

---

**Algorithm 1** Algorithm of `FedTextGrad`

**Input:** $N$ clients indexed by $i$, $B$: local mini-batch size, $C$: Client sampling rate. $T$: number of rounds

**Output:** Updated Prompts $P$

1: **ServerExecute**($C$):
2: Initialize $P^0$
3: **for** each round $t = 1, 2, \ldots T$ **do**
4:     $m \leftarrow \max(C \cdot N, 1)$
5:     $S_t \leftarrow$ (random set of $m$ clients)
6:     **for** each client $i \in S_t$ **in parallel do**
7:         $P_i^{t+1} \leftarrow$ ClientUpdate($i, P^t$)
8:     **end for**
9:     $P^{t+1} \leftarrow$ PromptAgg($[P_i^{t+1}]_{i \in S_t}$)
10: **end for**
11: Return Final $P^T$

12: **ClientUpdate**($i, P$):
13: $\mathcal{B} \leftarrow$ (Split $\mathcal{D}_i$ in to batches of size B)
14: **for** each local epoch $e = 1$ to $E$ **do**
15:     **for** each batch $b \in \mathcal{B}$ **do**
16:         $P \leftarrow$ TGD.step $\left(P, \frac{\partial \text{Evaluation}}{\partial P}\right)$
17:     **end for**
18: **end for**
19: Return $P$ to server.

---

the same batch is evaluated in a loop. The prompt is updated only if the performance does not drop compared to the previous non-updated version. Under homogeneous FL settings, each dataset is randomly split into 3 clients, each having an equal number of training and validation samples.

### 3.2 EMPIRICAL STUDY ON KEY HYPER-PARAMETER CHOICES

This section investigates the impact of key hyper-parameters, including local steps, number of clients and batch size, on `FedTextGrad` through ablation studies.

**Local steps**: Previous FL research (McMahan et al., 2017) has frequently conducted ablation studies on local steps to understand the balance between local model updates and global model synchronization. In traditional FL, increasing local steps is expected to reduce communication costs by allowing more local updates before synchronization with the server (McMahan et al., 2017; Li et al., 2020). However, this often comes at the cost of performance degradation due to local overfitting and divergence from the global model. As observed in Fig. 2a, increasing local steps in our setting leads to a

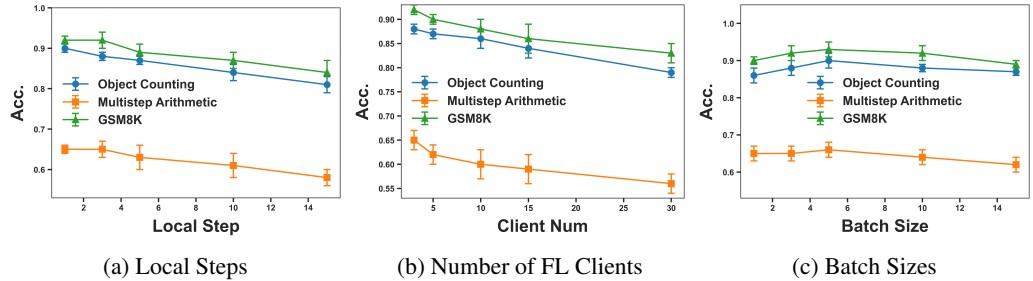

| (a) Local Steps | (b) Number of FL Clients | (c) Batch Sizes |

Figure 2: Ablation study of the impact of three key FL hyper-parameters on `FedTextGrad`, evaluated across three datasets.

significant performance drop, confirming that too many local updates without frequent synchronization exacerbate model misalignment across clients.

**Number of clients**: Previous work McMahan et al. (2017); Li et al. (2020) has explored the effect of increasing the number of clients in FL to evaluate the model's robustness to client heterogeneity and communication bottlenecks. In Fig. 2b, we examine this effect by splitting a single-task dataset into multiple subsets, each representing a client. With the increase in the number of clients, the performance drops dramatically. This can be attributed to communication overhead and misaligned prompt updates between the server and clients. Furthermore, in tasks like Object Counting, increasing the number of clients consistently degraded performance, likely due to the model's sensitivity to the heightened heterogeneity and divergent data distributions.

**Batch size**: Ablation studies on batch size in FL typically explore its impact on convergence and communication efficiency. Larger batch sizes are expected to stabilize training by reducing gradient variance, but they might also slow the convergence due to the reduced frequency of updates (McMahan et al., 2017). In Fig. 2c, it can be observed that increasing the batch size initially improves performance by smoothing the optimization process. After a certain threshold, performance declines. This is likely due to less frequent updates, which reduce the model's ability to adapt quickly to new data distributions, especially under distribution shifts.

**In summary**, our ablation studies reveal that while increasing local steps and batch size can initially stabilize and improve optimization, they ultimately introduce significant challenges related to communication efficiency and global model alignment. Similarly, increasing the number of clients improves performance up to a point, but leads to degradation due to communication and synchronization issues, particularly in data heterogeneous environments.

## 3.3 EVALUATION ON HETEROGENEOUS SETTINGS

**Heterogeneous Experimental Setup.** We evaluate the Heterogeneous `FedTextGrad` framework using three distinct datasets: Object Counting, Multistep Arithmetic, and GSM8K, with each dataset representing a client in the federated learning setting. The experiments focus on two critical hyperparameters: the number of local steps ($E$) and batch size ($B$). Local steps ($E$) refer to the number of client-specific updates performed before global model synchronization, while the batch size ($B$) determines the number of samples processed in each local update. To investigate the interaction between these hyperparameters, we conduct evaluations with $E \in \{3, 5, 10\}$ and $B \in \{1, 3, 10\}$. Each dataset is split evenly among clients, and the performance is assessed based on the model's ability to adapt under varying hyperparameter configurations.

Table 1: Performance of Heterogeneous `FedTextGrad` Framework Across Batch Sizes ($B$) and Local Steps ($E$). This table presents the performance of the Heterogeneous `FedTextGrad` framework using three datasets (Object Counting, Multistep Arithmetic, GSM8K) as clients in a federated learning setup, with a total of 3 clients.

| $E$ | $B = 1$ | $B = 3$ | $B = 10$ |
|---|---|---|---|
| 3 | 0.73 (0.03) | 0.78 (0.02) | 0.72 (0.03) |
| 5 | 0.83 (0.02) | 0.86 (0.02) | 0.84 (0.01) |
| 10 | 0.81 (0.03) | 0.83 (0.02) | 0.80 (0.02) |

**Results and Observations.** The results presented in Table 1 reveal notable patterns in the Heterogeneous `FedTextGrad` framework's performance across different local steps and batch sizes. Increasing local steps ($E$) from 3 to 5 improves performance across all batch sizes, indicating that more local updates allow clients to better capture their specific data distributions. However, further increasing $E$ to 10 leads to a slight performance degradation, suggesting that excessive local updates before synchronization may result in overfitting to client-specific data. Similarly, batch size ($B$) exhibits an optimal value at $B = 3$, which consistently delivers the best results. Larger batch sizes, such as $B = 10$, show diminishing returns or even performance drops, possibly due to the reduced frequency of updates, which hampers the model's ability to adapt effectively to local distributions. These findings underscore the importance of careful tuning of local steps and batch size to balance local adaptation with global model convergence.

### 3.4 PERFORMANCE WITH VARIOUS LLM APIs.

**Experimental Setup.** We evaluate the performance of various LLM application programming interfaces (APIs) on the BBH Object Counting dataset, considering both centralized and federated learning settings. In the federated learning setup, we use the homogeneous `FedTextGrad` configuration with the following default hyperparameters: local steps ($E = 3$), batch size ($B = 3$), and three clients, each receiving an evenly split portion of the dataset. For the centralized setting, the split datasets are grouped and trained as a single dataset, enabling a direct comparison between centralized learning and federated learning.

**Results.** The results, illustrated in Figure 3, highlight several notable trends. In the centralized TextGrad setting (Figure 3a), GPT-4 (Achiam et al., 2023) and DeepSeek R1 distill Llama 70B (Guo et al., 2025) model achieve the highest accuracy (0.99), closely followed by LLaMA-3.1-405B (0.96) and LLaMA-3.1-70B (0.95). In the federated learning setting (Fig. 3a), while GPT-4 continues to perform best (0.98), there is a slight performance drop across all models when transitioning from centralized to federated learning. The performance gap is more pronounced in smaller models, such as Gemma-2-9B (Team et al., 2024) and Qwen-2-7B (Yang et al., 2024a), which experience sharper declines in accuracy (see Fig. 3b). These findings suggest that while federated learning has a marginal effect on more capable models like GPT-4 and LLaMA, the impact is more substantial for less powerful LLMs, underscoring the challenges of federated learning in heterogeneous environments.

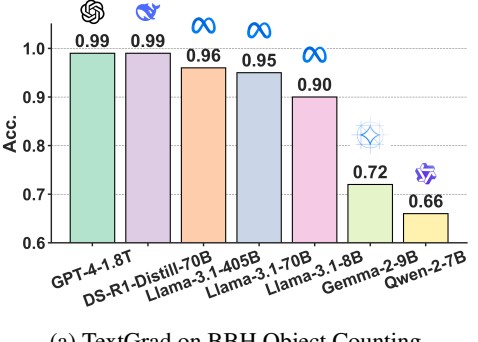
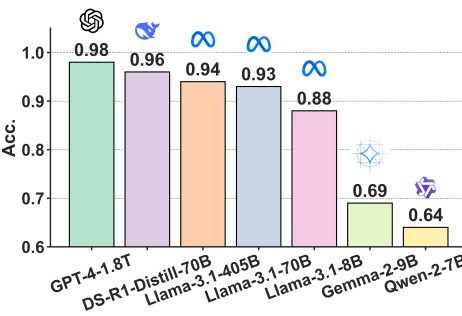

(a) TextGrad on BBH Object Counting          (b) `FedTextGrad` on BBH Object Counting

Figure 3: Comparison of the impact of different LLMs on (a) Centralized TextGrad and (b) `FedTextGrad` for BBH Object Counting tasks.

## 4 ENHANCED `FedTextGrad` PROMPT AGGREGATION METHOD

In the following section, we first highlight the limitations of directly applying prompt concatenation for prompt aggregation in `FedTextGrad`, demonstrating that this approach is impractical due to the excessive token length it generates, which often exceeds the context window of LLM API and leads to processing errors. Second, we explore summarization as an alternative method to mitigate this issue; however, we observe that it consistently underperforms compared to concatenation.

Finally, inspired by principles of human communication, we introduce an enhanced summarization approach that incorporates uniform information density. We find that this simple yet effective method significantly improves performance while maintaining practical token lengths.

## 4.1 PROMPT CONCATENATION ANALYSIS

**Concatenation aggregation in** `FedTextGrad` **risks exceeding token limits and increasing costs, posing scalability challenges.** Concatenation is a natural approach for aggregating text information; however, this method introduces a significant issue in our `FedTextGrad` framework, as the prompt length increases with the number of clients, potentially exceeding token limits and leading to rejection by LLM API services. The Fig. 4 illustrates the exponential growth in concatenated prompt token length as the number of clients increases, highlighting the risk of exceeding GPT-4's context window limit of 8192 tokens (denoted by the red dashed line). The right y-axis shows the associated cost in USD, with increasing token lengths resulting in higher costs. Error bars represent the standard deviation of token lengths across different client configurations. The figure emphasizes the trade-off between prompt length and scalability in federated learning settings, particularly when using concatenation-based prompt aggregation methods.

## 4.2 CONCATENATION VS. SUMMARIZATION

Summarization is often regarded as a natural solution to mitigate the issue of long token lengths introduced by concatenation. However, in our `FedTextGrad` framework, summarization underperforms compared to concatenation, prompting the need for further enhancements. In this section, we compare concatenation and summarization as prompt aggregation strategies. The results, shown in Fig. 6a, cover three tasks: *Object Counting*, *Multistep Arithmetic*, and *GSM8K*. For the *Object Counting* task, concatenation slightly outperforms summarization with accuracies of 0.90 and 0.88, respectively. In the more complex *Multistep Arithmetic* task, the performance gap is more pronounced, with concatenation (0.69) significantly surpassing summarization (0.55). In contrast, for the *GSM8K* task, both methods perform comparably, with concatenation

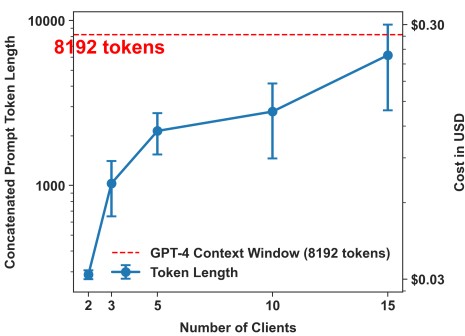

Figure 4: Increasing token length of concatenated prompts.

achieving 0.94 accuracy and summarization closely following at 0.92. Overall, concatenation consistently demonstrates superior performance, particularly in more complex tasks like multistep arithmetic, underscoring its advantage in our framework.

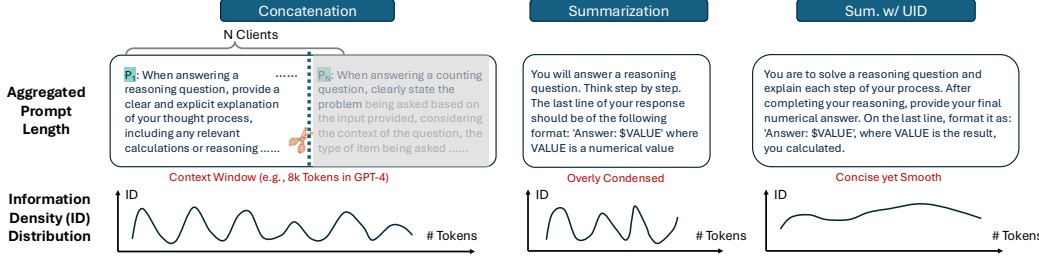

Figure 5: Illustration of the three types of prompt aggregation proposed in this paper: *1) Concatenation* – where prompts from clients are directly concatenated; *2) Summarization* – where a large language model (LLM) is employed to summarize the prompts provided by the clients; *3) Summarization with UID (SUM w/ UID)* – where the summarization process is enhanced by applying uniform information density principles.

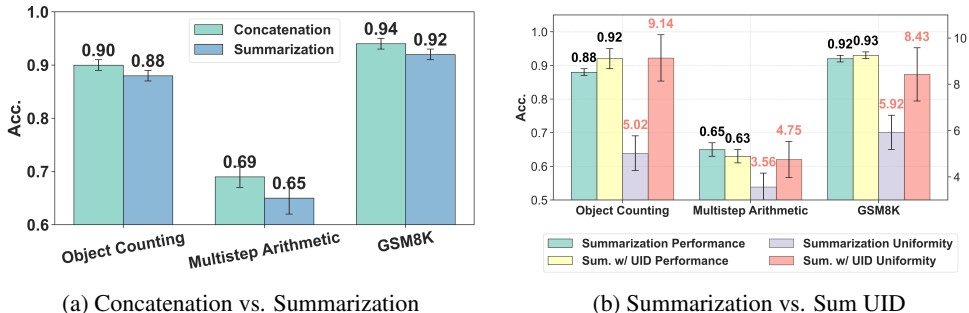

(a) Concatenation vs. Summarization      (b) Summarization vs. Sum UID

Figure 6: Observations of different aggregation strategies, where (a) presents the performance of concatenation and summarization, and (b) compares the performance of *UID* and summarization.

### 4.3 THE PROPOSED SUM *UID* PROMPT AGGREGATION METHOD

**We propose a UID-based prompt summarization method to overcome the limitations of summarization's information non-uniformity, enhancing the stability and accuracy of** `FedTextGrad`. Fig. 5 highlights the effects of different prompt aggregation approaches. The main challenge lies in efficiently combining local prompt updates into a global prompt that retains essential information while adhering to input length constraints for federated optimization. Direct concatenation of client prompts can produce overly long global prompts, particularly with many FL clients, potentially exceeding the LLM's context window. Summarization addresses this issue by keeping the global prompt within the allowed length, but it often creates overly dense prompts that degrade model performance.

**UID Hypothesis.** To address the issue of excessive token lengths in concatenation, we propose a prompt aggregation method based on the **Uniform Information Density Hypothesis (UIDH)** (Meister et al., 2020), which posits that effective communication involves distributing information uniformly. We hypothesize that uneven information distribution in prompts adversely affects LLM performance, as critical updates from clients may be diluted. To mitigate this, we introduce an enhanced summarization approach incorporating **UID principles** to ensure a balanced representation of client updates in the aggregated prompt.

**Measuring Uniformity.** To measure information density uniformity (Meister et al., 2020), surprisal values are computed for each word in a text based on the conditional probabilities derived from a pre-trained language model, such as GPT-2. The uniformity is quantified by the variance in surprisal values, where lower variance indicates a more uniform distribution of information. The process involves tokenizing the text, extracting log-probabilities, calculating surprisal for each token, and then computing the mean and variance of these values. The mean surprisal ($\mu$) represents the average information density, while the variance ($\sigma^2$) reflects the uniformity of information distribution: $\mu = \frac{1}{N}\sum_{i=1}^{N} I(w_i|C)$, $\quad \sigma^2 = \frac{1}{N}\sum_{i=1}^{N}(I(w_i|C) - \mu)^2$ where $I(w_i|C) = -\log_2 P(w_i|C)$ is the surprisal for token $w_i$ given context $C$, and $N$ is the number of tokens in the text. A lower value of $\sigma^2$ indicates higher uniformity in information density, consistent with the *Uniform Information Density* hypothesis.

**Performance Gains.** Our method improves prompt aggregation by maintaining a uniform information density across the summarized prompt, preserving key information from each client while preventing over-compression. Fig. 6a shows that, compared to summarization, our UID-based method yields superior performance in Object Counting and GSM8K dataset. Our empirical results demonstrate consistent gains in accuracy and prompt stability, confirming the effectiveness of applying UID principles in FL systems.

## 5 AN ENVISIONED ROADMAP FORWARD

With LLMs becoming a burgeoning field and model sizes continuously increasing, addressing the high costs and data privacy concerns in LLM training is paramount. Adapting FL to LLMs offers a promising direction. `FedTextGrad` introduces a novel and efficient paradigm that utilizes LLMs as optimizers and textual gradients to update LLM components. However, practical implementation of `FedTextGrad` involves several critical challenges, including (1) managing heterogeneous data

to prevent conflicting contexts and ensure effective aggregation, (2) developing privacy-preserving methods tailored to textual gradients, as traditional approaches often compromise utility in natural language settings, (3) improving communication efficiency through advanced summarization and adaptive encoding to scale in federated environments, and (4) ensuring robustness against diverse adversarial attacks by adapting different defense mechanisms.

**Learning on Heterogeneous Environment:** When encountering heterogeneous data and model architecture, clients can produce conflicting contexts, which result in failed texture aggregation or ambiguous summarized texts. In traditional federated learning (FL), strategies such as resolving gradient conflicts with flatter minima (Chen et al., 2023) or regularizing clients' gradient updates (Li et al., 2020), as well as sharing common hidden features (Yi et al., 2024), have proven effective. However, these approaches depend on numerical operations or the use of hidden features, making them unsuitable for scenarios involving textual gradients or black-box settings. Consequently, adapting existing methods or developing novel solutions represents a crucial direction for future research.

**Privacy Attack and Protection**: By sharing texture gradient from client LLMs, `FedTextGrad` exposure attack surfaces. Although direct gradient inversion might be less effective on textual gradients, adversaries could still use the contextual nature of text to uncover private information, making careless prompt engineering a significant risk for revealing sensitive data (Yao et al., 2024). In traditional FL, privacy protection methods such as differential privacy (DP), secure multi-party computation (SMPC), and homomorphic encryption are widely used to safeguard numerical gradients (Behnia et al., 2022). However, these strategies face challenges when applied to federated textual gradients, as natural language contains more context and meaning, making it harder to obfuscate without losing utility. While DP could introduce noise into text, this risks rendering the gradients incoherent, and SMPC or encryption techniques would require significant advances to handle the complexity of encrypted text. Thus, new privacy-preserving methods tailored specifically for textual data are needed in federated learning.

**Communication Efficiency**: Traditional FL approaches to improving communication efficiency typically employ techniques such as gradient compression (Jiang et al., 2022) or conflict resolution to reduce the number of communication rounds (Chen et al.). However, none of these methods are specifically designed for the textual gradient domain. Unlike numerical gradients, textual gradients are inherently contextual and carry semantic information, making direct compression or sparsification infeasible without risking the loss of critical information or coherence, highlighting the need for further research into scalability within `FedTextGrad`.

**Robustness**: Research in this area examines various methods, such as poisoning attacks, Byzantine attacks, and other empirical approaches, that adversaries use to undermine the integrity of global models in FL systems. To counter these attacks, various defense mechanisms have been developed to enhance robustness in FL. For example, Byzantine-resilient aggregation methods like Krum and Trimmed Mean mitigate malicious updates by focusing on reliable client contributions and have been widely adopted in traditional FL (So et al., 2020; Jin & Li, 2023) but infeasible in the textural context. Methods based on outlier detection can identify and remove suspicious updates. With proper prompt design or text embedding, such strategies show potential for use in `FedTextGrad`. However, the inherent complexity of LLM-based systems exacerbates the difficulty of both executing these attacks and defending against them.

## 6 CONCLUSIONS

In this work, we introduced `FedTextGrad`, an extension of the TextGrad framework specifically designed to address the challenges of prompt optimization in federated learning settings. By identifying the training instability caused by aggregating distributed prompt updates and demonstrating the limitations of traditional concatenation and summarization-based techniques, we proposed a novel approach based on Uniform Information Density Principles to enhance `FedTextGrad` prompt summarization. Our method addresses the issue of uneven information distribution, leading to improved prompt efficacy and overall performance in federated environments. This study establishes a foundation for future advancements in prompt optimization for large-scale, distributed learning systems and opens new avenues for deploying LLMs in privacy-sensitive, resource-constrained environments.

ACKNOWLEDGEMENTS

The research is supported by Natural Science and Engineering Research Council of Canada (NSERC), Canada CIFAR AI Chairs program, Canada Research Chair program, MITACS-CIFAR Catalyst Grant Program, the Digital Research Alliance of Canada. The research is also supported, in part, by the Ministry of Education, Singapore, under its Academic Research Fund Tier 1; and the National Research Foundation, Singapore and DSO National Laboratories under the AI Singapore Programme (AISG Award No. AISG2-RP-2020-019).

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

# A    RELATED WORK

## A.1    FEDERATED LEARNING WITH LLMS

**Federated Learning.**    FL is a privacy-preserving collaborative training paradigm that allows multiple parties to build a shared global model without the need to exchange raw data. One of the main challenges in FL is data heterogeneity, as client datasets often stem from different distributions (McMahan et al., 2017). To address this issue, various techniques have been proposed, including regularization (Li et al., 2020), gradient correction (Niu & Deng, 2022), feature alignment (Yu et al., 2021), adaptive aggregation weights (Wu & Wang, 2021), momentum introduction (Liu et al., 2020), and leveraging pre-trained models (Huang et al., 2023).

**Fedeated Learning with LLMs.**    As LLMs have achieved significant success in centralized learning, there is a growing interest in adapting FL to accommodate the fine-tuning of pre-trained LLMs (Ren et al., 2024), particularly to supplement the publicly available data with privately owned datasets (Jin & Li, 2023). In response, several frameworks have emerged recently, including OpenFedLLM (Ye et al., 2024) and FederatedScope-LLM (Kuang et al., 2024). Moreover, advanced methods such as FedbiOT (Wu et al., 2024) which safeguards model ownership, and FFA-LoRA (Sun et al., 2024) which enhances performance under differential privacy constraints, are being developed to optimize LLM training in federated environments.

**Privacy in LLM Prompting.**    LLM prompting have revolutionized natural language processing but face significant challenges when handling privacy-sensitive text data, a topic that remains relatively underexplored. Chong et al. (2024) propose "Casper," a browser extension that sanitizes user prompts by removing sensitive information before submission to LLMs. Edemacu & Wu (2024) survey privacy-preserving prompt engineering techniques, emphasizing approaches such as differential privacy and data obfuscation. Gim et al. (2024) introduce "Confidential Prompting," which leverages confidential computing to secure user prompts during LLM inference. Yu et al. (2024) tackle privacy concerns in instruction tuning by generating synthetic instructions under differential privacy guarantees, reducing data exposure risks. Finally, Li et al. (2024) present "LLM-PBE," a toolkit designed to evaluate privacy risks and mitigation strategies in LLMs. These studies highlight the pressing need for robust privacy-preserving mechanisms in LLM prompting applications. Building on these advancements, future work can explore integrating privacy-preserving techniques, such as differential privacy, data obfuscation, or confidential computing, into FedTextGrad to secure prompts or textual gradients and mitigate privacy risks in federated learning scenarios.

## A.2    LLMS AS OPTIMIZERS

**Prompt Optimization.**    Prompt optimization has attracted significant attention, with various strategies proving effective in enhancing the performance of LLMs. Techniques such as selecting optimal few-shot examples (Pryzant et al., 2023), in-context learning (Dong et al., 2022), chain of thought reasoning (Wei et al., 2022), and model ensembles (Jiang et al., 2023) have shown promise. Furthermore, several strategies have been developed to automate this process. White-box approaches, which rely on numerical gradients, offer a useful solution. However, they are limited by the need to access model parameters, restricting their applicability to only open-source LLMs.

**LLMs as Optimizier.**    Recent research has turned towards leveraging *LLMs as optimizers* in black-box settings (Yang et al., 2023). The foundation of this concept stems from the ability of LLMs to simulate human decision-making. Zheng et al. (2023) benchmarked the behavior of LLMs and human decisions, finding that modern LLMs align closely with human judgment. Building on this, Yang et al. (2024b) proposed *optimization by prompting*, where LLMs generate new solutions based on a prompt that includes previously generated solutions. Ma et al. (2024) further investigated whether LLMs are effective prompt optimizers. Tools like DSPy Khattab et al. (2023) and ProTeGi Pryzant et al. (2023) introduced programmatic frameworks for optimizing LLM-based APIs, achieving performance gains across tasks such as question answering and prompt refinement. All LLM-as-optimizer approaches require the LLM to be as powerful (large-scale) as possible, as smaller LLMs currently lack the capability to serve as effective optimizers. However, it is feasible

to reuse prompts optimized by large-scale models for smaller ones Vu et al. (2022), enabling the adaptation of the LLM-as-optimizer paradigm in resource-constrained settings

**TextGrad.** Recently, TextGrad (Yuksekgonul et al., 2024) presents a more generalized approach of using LLM as optimizers by adapting the above ideas to broader domains, such as optimizing instances like molecular structures or code snippets, using a *textual-backpropagation-based* framework. These methods highlight the versatility of LLMs in enhancing their own outputs across diverse applications, thus opening up opportunities for prompt optimization in closed-source LLMs within centralized learning settings by circumventing the need for access to model parameters. However, the research question of how to achieve similar advances in FL settings remains unresolved. This paper seeks to address this question. Nevertheless, adapting FedTextGrad to resource-constrained settings with smaller LLMs remains a challenging and unresolved research question, as smaller LLMs often lack the capacity to serve effectively as LLM-as-optimizers for self-refinement.

## B    EXPERIMENTAL DETAILS

### B.1    DATASETS

We evaluate our method on three primary reasoning tasks:

- **BBH Object Counting** (Srivastava et al., 2022): A task challenges models to accurately count objects based on visual or textual descriptions, testing their ability to reason about quantities and manage multiple elements within a scene or context.
- **BBH Multi-Step Arithmetic** (Srivastava et al., 2022): Another BBH task that tests a model's ability to solve mathematical problems that require multiple sequential steps of reasoning, assessing its proficiency in handling complex, multi-stage arithmetic operations.
- **GSM8k Math Problem** (Cobbe et al., 2021): A dataset of grade school math problems designed to test the mathematical reasoning capabilities of LLMs.

### B.2    BASE MODELS

We conduct experiments using five large language models (LLMs), encompassing both widely-used commercial APIs such as *GPT-4o* and *GPT-3.5* (Achiam et al., 2023), as well as cutting-edge open-source models like *Llama 3*, *Llama 3.1* (Dubey et al., 2024), and *Qwen 2* (Yang et al., 2024a).

By leveraging this diverse set of models, we are able to rigorously assess the scalability and robustness of our approach across a range of architectures and model sizes, ensuring comprehensive evaluation and applicability.

### B.3    `FedTextGrad` SETUP

All datasets are split into **training**, **validation**, and **test** sets. The training set is used to optimize prompts through `FedTextGrad`, the validation set helps with the prompt selection and hyper-parameter tuning, and the test set is reserved for reporting the final performance. FL simulates a decentralized setting where clients send prompt updates to a central server without sharing raw data.

## C    PROMPT AGGREGATION TECHNIQUES

Prompt aggregation plays a critical role in federated learning with LLMs, especially as the number of clients increases. We explore two primary methods for aggregating client prompts and evaluate their effectiveness under varying conditions.

- **Concatenation**: In this method, the individual prompts from each client are concatenated into a single, aggregated prompt. While simple to implement, this approach has significant drawbacks. As the number of clients increases, the total prompt length can easily exceed the input length constraints imposed by large language models (LLMs), such as GPT-4's context window of 8192 tokens. This results in prompts being truncated or rejected by the LLM API, severely limiting the scalability of this approach in federated settings.

- **Summarization**: To alleviate the issue of prompt length in concatenation, summarization techniques are applied to compress the information from each client into a shorter prompt. Although this reduces token length, it often comes at the cost of performance degradation. The compression inherent in summarization leads to information loss, particularly when the prompts contain complex or diverse client-specific updates. This information loss can cause suboptimal model performance, especially in tasks that require retaining detailed and nuanced client data.

- **Summarization with Uniform Information Density**: We introduce a summarization approach based on the Uniform Information Density (UID) hypothesis, which ensures a more balanced distribution of information within aggregated prompts. The UID hypothesis suggests that distributing information uniformly optimizes communication efficiency, and we apply this principle to mitigate the performance degradation observed in traditional summarization methods. By maintaining uniform information density, our method preserves critical information from each client while reducing prompt length, aligning with LLM input constraints. This approach consistently enhances performance across tasks by improving reasoning accuracy and prompt stability in federated learning environments.

**An Example of the Prompt Designed for Summarizing Prompts From TextGrad.**

> **Prompt for Summarization**
>
> ```
> Merge the following list of prompts into a single, cohesive
> prompt while preserving all original information.  Ensure
> that the final instruction remains unchanged and is placed
> as the last sentence.  Provide only the merged prompt.
> ```

**An Example of the Prompt Designed for Summarizing Prompts with UID From TextGrad.**

> **Prompt for Summarization with UID**
>
> ```
> Merge the following list of prompts into a single, cohesive
> prompt while preserving all original information.  Apply
> Uniform Information Density Principles.  Ensure that the
> final instruction remains unchanged and is placed as the
> last sentence.  Provide only the merged prompt.
> ```

**A Concatenated Prompt Example.**

---

**Concatenated Prompt for Object Counting**

When answering a reasoning question, provide a clear and explicit explanation of your thought process, including any relevant calculations or reasoning, to support your answer. Use specific language to explain your reasoning, and avoid using vague or ambiguous language that could be interpreted in multiple ways.

Provide a detailed and step-by-step explanation of your thought process, including any relevant calculations or reasoning, to support your answer. For example, a good response might be: 'To determine the total number of objects, I counted each item individually: 1 microwave, 1 table, 1 fridge, 1 stove, 1 oven, 1 toaster, 1 couch, and 4 cars. Therefore, the total number of objects is 1 + 1 + 1 + 1 + 1 + 1 + 1 + 4 = 11.'

Ensure that your response is clear, concise, and free of unnecessary words or phrases, and that it clearly addresses the question being asked. Use precise and descriptive language to explain your reasoning, avoiding oversimplification and providing a nuanced or detailed explanation of the answer.

Consider the potential for ambiguity in your response and avoid using language that could be interpreted in multiple ways. Provide a clear and concise explanation of the reasoning behind your answer, using relevant details and examples to support your response.

When providing a numerical answer, avoid including unnecessary phrases or context, and focus on presenting the answer in a clear and concise format, such as a single number or a brief explanation of the calculation.

When answering a counting question, provide a clear and explicit statement of the count, using a specific format such as 'Answer: X' where X is the numerical value. When providing a final answer, explicitly state the operations performed to arrive at the answer, and provide a clear and concise explanation of the reasoning behind the answer.

Use precise and descriptive language to explain your reasoning, avoiding oversimplification and providing a nuanced or detailed explanation of the answer. Address any missing information in the problem and provide a complete and accurate response.

Use specific details and concrete examples to support your response and provide a clear and concise explanation of the reasoning behind the answer. Provide a final answer that explicitly states the operations performed to arrive at the answer and includes a clear and concise explanation of the reasoning behind the answer.

Use a formal and objective tone to ensure that the response is clear and unambiguous. If ambiguity is unavoidable, provide a clear and concise explanation of the ambiguity, and ensure that the response is still clear and unambiguous. Ensure that the response directly addresses the question being asked and provides a clear and concise answer to the problem.

**A Summarized Prompt Example.**

---

**Summarized Prompt for Object Counting**

```
When answering a counting question, clearly state the
problem being asked based on the input provided, considering
the context of the question, the type of item being asked
about, and any relevant information that may affect the
answer.

Identify and count the specific category of items mentioned
in the question, and provide a clear and concise count of
the objects mentioned.  Ensure that the numerical answer
is accurate and precise, and provide a clear and concise
step-by-step explanation of how you arrived at your answer.

Be aware of idiomatic and colloquial language that may
affect the answer, and use your best judgment to interpret
any unclear or ambiguous language in the question.  Consider
alternative scenarios or edge cases that may affect the
answer, and use relationship understanding to disambiguate
any unclear or ambiguous language in the question.

Provide a direct answer to the problem being asked, in the
format "The total number of [object type] is [number]".
Avoid paraphrasing the input and use step-by-step
explanations to provide a clear understanding of the
calculation or reasoning behind the answer.

Specify the type of objects being counted, such as
'animals,' 'fruits,' or 'household items,' based on the
input provided.  To ensure accuracy, please count each
type of object individually and add them together.  For
example, if the question asks for the total number of
musical instruments, count each type separately (e.g.,
guitars, violins, drums).

Answer:  $VALUE where VALUE is a numerical value.  The last
line of your response should be of the following format:
"Answer:  $VALUE" where VALUE is a numerical value.
```

---

## D  EXPERIMENTS ON MORE CHALLENGING AND HIGH-COMPLEXITY TASKS WITH FEDTEXTGRAD

To evaluate the performance of FedTextGrad on tasks with higher complexity and reasoning challenges, we conducted experiments using GPT-4o on datasets extracted from *LiveBench* (White et al., 2024). These datasets include tasks that test logical inference, spatial reasoning, and mathematical abstraction, providing a rigorous benchmark for assessing the robustness of our FedTextGrad.

**Experimental Setup.**  We evaluated FedTextGrad on two categories of tasks: reasoning and advanced mathematical problems. For reasoning tasks, we utilized *Web of Lies (Version 2)*, an enhanced dataset that introduces deductive red herrings to challenge logical rigor; *Zebra Puzzle*, a deductive reasoning task involving multiple constraints across variables like colors and nationalities; and a *Spatial Dataset*, requiring the model to reason about numerical and positional attributes of solid, regular heptagons. For mathematical tasks, we employed the *AMPS Hard Dataset*, designed to test advanced symbolic manipulation and mathematical reasoning through challenging, randomized problem distributions.

Table 2: Performance of Centralized and Federated Configurations on Reasoning and Mathematical Tasks. Best test accuracies are reported for each method across the datasets. Results for Centralized `TextGrad` and `FedTextGrad` with summarization.

| Category | Dataset | Centralized TextGrad | FedTextGrad |
|---|---|---|---|
| **Reasoning** | Spatial | 0.53 | 0.40 |
| | Web of Lies | 0.37 | 0.30 |
| | Zebra Puzzle | 0.33 | 0.27 |
| **Math** | AMPS Hard | 0.46 | 0.50 |

Table 3: Results of Prompt Transferability from LLaMA 3.2-11B to LLaMA 3.2-3B. The table reports the performance of prompts optimized on the larger model and directly transferred to the smaller model, compared to initial prompts without optimization. Performance metrics are prediction accuracy on the task (higher is better).

| Task | Initial Prompt ↑ | Transferred Prompt ↑ | Performance Change |
|---|---|---|---|
| Object Counting | 0.66 | **0.69** | +0.03 |
| Multi-step Arithmetic | 0.51 | **0.66** | +0.15 |
| GSM8K | **0.80** | 0.72 | -0.08 |

**Results.** In Table 2, on reasoning tasks, FedTextGrad with summarization demonstrates adequate performance but remains below the centralized TextGrad configuration across all datasets. For the *Web of Lies* dataset, FedTextGrad achieves an accuracy of 0.30, which is lower than the centralized TextGrad accuracy of 0.37, highlighting challenges in adapting to deductive reasoning tasks in federated settings. Similarly, on the *Zebra Puzzle* dataset, FedTextGrad achieves 0.27 compared to the centralized TextGrad's 0.33, reflecting the difficulty of effectively optimizing logical reasoning tasks in a decentralized environment. For the *Spatial* dataset, FedTextGrad records 0.40 accuracy compared to the centralized TextGrad's 0.53, further showcasing challenges in handling spatial reasoning under federated conditions.

In contrast, results for the mathematical task (*AMPS Hard Dataset*) show that FedTextGrad surpasses centralized TextGrad with an accuracy of 0.50 versus 0.46. This indicates that, despite challenges in reasoning tasks, FedTextGrad excels in tasks requiring mathematical reasoning, possibly due to its ability to better capture client-specific variations in structured numerical tasks. These results underscore the varying effectiveness of FedTextGrad across different task domains, with potential for further improvements in reasoning tasks under federated settings.

# E    FEASIBILITY OF SMALLER LLMS DEPLOYMENT WITH TEXTGRAD.

**Experimental Setup.** To evaluate the feasibility of deploying prompts optimized on larger LLMs in resource-constrained settings, we conducted an additional experiment on *prompt transferring*. Specifically, prompts optimized using TextGrad on a larger model (`LLaMA 3.2-11B`) were directly applied to a smaller model (`LLaMA 3.2-3B`) without further optimization. Unlike the `LLaMA 3.1-8B` model used in the main text, we opted for the `LLaMA 3.2` series because the `3.1` series does not include models smaller than 8B. The `LLaMA 3.2` series, on the other hand, offers a wider range of model sizes, making it suitable for evaluating prompt transferability across different model scales. The tasks used were the same as in the main text, including *BBH Object Counting*, *BBH Multi-step Arithmetic*, and *GSM8K*.

**Results.** The results in Table 3 showed that on *BBH Object Counting* and *BBH Multi-step Arithmetic*, the transferred prompts achieved significantly better performance compared to the initial prompts. This indicates the potential for reusing optimized prompts in smaller, resource-efficient models without requiring further optimization. However, on the *GSM8K* task, the transferred prompts experienced a noticeable performance downgrade. This highlights the challenges of prompt

Table 4: Averaged performance with client heterogeneity of summarization and UID summarization across three clients under varying batch sizes ($B$). Performance is reported as the mean accuracy across tasks.

| Method | B = 1 | B = 3 | B = 10 |
|---|---|---|---|
| Summarization | 0.73 (0.03) | 0.78 (0.02) | 0.72 (0.03) |
| UID Summarization | **0.75 (0.02)** | **0.79 (0.02)** | **0.74 (0.03)** |

generalization for more complex reasoning tasks. These findings suggest that prompt transferring is a promising approach for leveraging the optimization capabilities of larger LLMs while deploying smaller models in resource-constrained settings. Nevertheless, the observed limitations in tasks like *GSM8K* underscore the need for further studies to enhance the generalization capabilities of prompt transferring within the TextGrad framework. This represents an important direction for future research.

## F UID SUMMARIZATION GENERALIZABILITY ON CLIENT HETEROGENEITY.

**Experimental Setup.** We evaluate the robustness of UID-based summarization in heterogeneous client settings, where each client is assigned a distinct task. This configuration follows the setup described in Section 3.3 of the main text. The tasks used to simulate client heterogeneity include reasoning-based benchmarks, and the model employed is `LLaMA 3.1-8B`.

In this setup, three clients handle unique tasks to represent heterogeneity in data distributions. Specifically, one client addresses object counting tasks, another manages multi-step arithmetic, and the third tackles problems from GSM8K. UID-based summarization is compared with standard summarization aggregation to measure its performance under these conditions. Additionally, performance trends are analyzed as the batch size increases to better understand the behavior of the summarization methods. The training epoch is fixed to 3 as the default for all experiments.

**Results** The results in Table 4 demonstrate that UID-based summarization performs effectively under moderate client heterogeneity. It achieves slightly better results than standard summarization aggregation, showcasing its ability to retain essential information while adapting to diverse data distributions. Moreover, the performance trends with increasing batch size closely mirror those observed for standard summarization, indicating consistent and stable behavior across different configurations. These findings highlight the effectiveness of UID summarization in federated settings with heterogeneous client data. They reinforce its applicability in real-world scenarios, where client heterogeneity is a common challenge. The additional results and analysis are included in the revised manuscript to provide a comprehensive evaluation of the method under varying conditions.

## G DYNAMIC PROMPT AGGREGATION

To explore the feasibility of dynamic aggregation strategies in federated learning, we conduct an experiment evaluating the performance of dynamic switching between concatenation and summarization during the aggregation stage. This experiment extends the analysis presented in Figure 6(a), using the same dataset and `LLaMA 3.1-8B` model.

**Experimental Setup.** In this experiment, we implement a dynamic aggregation strategy where concatenation is used for prompt aggregation by default. However, if the concatenated prompt exceeds the model's pre-defined context window (selected as 800 tokens), summarization is applied to reduce the prompt length. This hybrid approach leverages the strengths of both methods, aiming to balance information retention and context limitations. The federated setup follows the same configuration as described in Figure 6(a) of the main text.

**Results.** The results, shown in Figure 7, demonstrate that the dynamic aggregation strategy underperforms compared to both the summarization-only and concatenation approaches in general.

Notably, in one task (object counting), the dynamic strategy slightly outperforms summarization. However, in other tasks, its performance lags behind. These findings highlight the potential adaptability of dynamic aggregation switching in managing varying prompt lengths.

**Discussion.** This experiment underscores the promise of dynamic aggregation switching as an effective strategy for federated textual aggregation. By addressing context window constraints dynamically, the method balances the trade-off between information retention and prompt length management. However, a significant limitation is the need to pre-select an optimal context window for specific datasets, which can be challenging and requires dedicated selection and prior knowledge. Future work could focus on refining the switching criteria and exploring its applicability across a broader range of datasets and models.

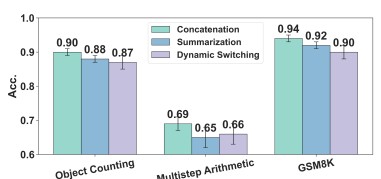

Figure 7: Concatenation vs Summarization vs Dynamic Aggregation Performance.

## H    SURPRISAL ANALYSIS ON SUMMARIZATION WITH AND WITHOUT UID

To investigate the information retention capabilities of UID summarization compared to standard summarization, we conduct an experiment analyzing the mean and maximum surprisal values of prompts after aggregation. Surprisal values measure the unexpectedness of generated text, providing insights into the uniformity and completeness of information across aggregated prompts.

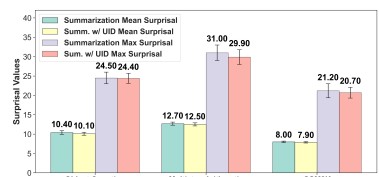

Figure 8: Summarization vs UID Summarization Aggregation Prompt Surprisal Value.

**Experimental Setup.** We calculate the mean and maximum surprisal values of prompts aggregated using both standard summarization and UID summarization. The calculations are performed across multiple tasks to ensure a comprehensive evaluation. Surprisal values are derived from the aggregated prompts post-processing, capturing how effectively the summarization methods retain critical information.

**Results.** The results in Figure 8 indicate that the mean and maximum surprisal values for UID summarization are nearly identical to those for standard summarization across all tasks. This suggests that both methods exhibit similar levels of information retention. While UID summarization is specifically designed to enhance information uniformity, it does not introduce additional information loss compared to standard summarization, as evidenced by the surprisal metrics.

**Discussion.** These findings confirm that UID summarization retains essential information as effectively as standard summarization while achieving improved task performance, as demonstrated in Section 4. The analysis highlights the robustness of UID summarization in ensuring both critical information retention and enhanced uniformity.

