# OpenReview forum: "Can Textual Gradient Work in Federated Learning?"
_ICLR.cc/2025/Conference — ICLR 2025 Poster_

### Official Review · Reviewer_hPgp · 2024-11-01

**Soundness:** 2
**Presentation:** 3
**Contribution:** 2
**Rating:** 6
**Confidence:** 3

**Summary:**

The paper presents FedTextGrad, a novel federated learning approach that incorporates textual gradients, allowing for the optimization of Large Language Models in decentralized environments. It introduces a new aggregation method based on the Uniform Information Density principle to address the challenge of retaining essential information from distributed prompt updates, thereby enhancing the applicability of federated learning to text-based optimization.

**Strengths:**

(1) Originality: The originality of the paper lies in its pioneering effort to adapt TextGrad, a method for automatic differentiation via text, into the federated learning (FL) framework. By proposing FedTextGrad, the authors have expanded the scope of FL to leverage the capabilities of LLMs in a decentralized manner, without the need for explicit numerical loss functions.
(2) Quality: The paper provides a thorough analysis of the proposed method's feasibility and identifies key challenges associated with textual gradient aggregation in a federated setting.
(3) Clarity: The paper is clear and well-structured.
(4) Significance: The significance of the paper lies in its integration of textual gradients into the federated learning framework.

**Weaknesses:**

(1) two typographical errors was identified on lines 191 and 216 of the paper;
(2) The description of "Client" in the paper is unclear: Figure 2(b) shows "Client Num" exceeding three, but on line 312, the number of clients is stated to correspond to the number of datasets used in the paper, yet the paper only mentions three datasets;
(3) This paper introduces TextGrad into the federated learning framework, but experiments are only conducted on reasoning tasks, which is not sufficient in terms of experimental scenarios.

**Questions:**

(1) The paper, while thorough in its experimental section, could be further strengthened by incorporating a wider array of datasets and tasks, including but not limited to coding, problem-solving, medicine and chemistry, to enhance the generalizability of its findings.
(2) The paper could benefit from a more direct comparison with existing federated learning methods, especially those tailored for LLMs.
(3) Why was the Llama-3.1-8B model exclusively utilized for prompt optimization and ablation studies, and not a variety of models?

---

> ### Author Response · Authors · 2024-11-25
>
> **W1: Typographical Errors: Errors found on lines 191 and 216**.
>
> Thank you for pointing out the typographical errors on **Line 191-216**. We have corrected them in the revised manuscript.
>
> ---
>
> **W2: Clarification on Client Descriptions: Figure 2(b) shows “Client Num” > 3, but line 312 links clients to datasets, and only three datasets are mentioned**.
>
> Thank you for pointing out this ambiguity. To clarify:
>
> **Figure 2(b)**: This corresponds to the setup in Section 3.2, where we evaluated the impact of varying client numbers by splitting a single dataset into multiple subsets, representing different clients.
>
> **Line 312 and Section 3.3**: These refer to task heterogeneity experiments, where three datasets correspond to three distinct clients.
>
> We revised the manuscript to explicitly differentiate these contexts and ensure consistent terminology. Thank you for your valuable feedback.

---

> ### Author Response · Authors · 2024-11-25
>
> **W3/Q1: Justification on Experimental Scope**
>
> We appreciate the opportunity to clarify the rationale for focusing on reasoning tasks in our experiments and to clarify that the tasks under reasoning are already diverse. We also address the broader applicability of FedTextGrad.
>
> **1. Wide Applicability of Reasoning Tasks**:
> **Reasoning in LLMs** involves tasks like logical deduction, commonsense reasoning, and multi-hop question answering, requiring models to **go beyond pattern recognition to manipulate and infer complex information** [1]. Evaluating LLMs on reasoning tasks provides a comprehensive assessment of their capabilities and helps identify areas for targeted improvements, making it a **cornerstone of advancing LLM**. In our study, we evaluated FedTextGrad using diverse reasoning benchmarks, including **BBH Object Counting**, **Multi-step Arithmetic**, and **GSM8K**, which **encompass complex and varied scenarios**. These tasks serve as representative examples of real-world applications where textual gradients are highly impactful.
>
> **2. Reasoning Tasks Are Core to TextGrad and Highly Relevant to FL Setting**:
> **Reasoning Tasks Align with TextGrad’s Paradigm**. Reasoning tasks are particularly well-suited for evaluating TextGrad because they rely on step-by-step thought processes and can be optimized through iterative textual feedback and self-refinement—core elements of the TextGrad paradigm. This alignment is highlighted in Section 1 (see **Line 59 to 63**) and validated by the findings in the original TextGrad paper. Reasoning is actively-studied and complicated problems in LLM research.
>
> **Reasoning as a Key Challenge in LLM-based TextGrad Research**. Reasoning tasks also represent a significant area of active research in LLM development, as they are both challenging and crucial for applications requiring complex decision-making and understanding. By focusing on reasoning tasks, we ensure that TextGrad is tested in scenarios that reflect real-world FL applications, where iterative optimization and feedback mechanisms are essential.
>
> **Justification of Task Selection from TextGrad.** While the original TextGrad study included tasks such as coding, chemistry, and radiotherapy treatment, these were not included here for specific reasons: coding tasks involve single-prompt optimization, which differs fundamentally from federated learning’s collaborative framework, and adapting them would require significant modifications outside our scope; chemistry and radiotherapy datasets are not clearly publicly accessible, and their integration into FL would require specialized adjustments for domain-specific constraints and task heterogeneity. Our focus on reasoning tasks provides a generalizable and accessible benchmark for evaluating TextGrad in FL scenarios.
>
> **3. Expanded Scope in Revised Version**:
> While reasoning tasks remain central to our study, we recognize the importance of exploring additional scenarios to demonstrate FedTextGrad’s versatility. In response to this feedback, we have added experiments on diverse tasks, diverse and challenging tasks from **LiveBench** [2] in **Appendix D**, in the revised version. These results showcase the broader applicability of FedTextGrad while reinforcing its effectiveness in reasoning-centric applications.
>
> | **Category**    | **Dataset**      | **Centralized TextGrad** | **FedTextGrad** |
> |------------------|------------------|---------------------------|------------------|
> | **Reasoning**    | Spatial          | 0.53                      | 0.40            |
> |                  | Web of Lies      | 0.37                      | 0.30            |
> |                  | Zebra Puzzle     | 0.33                      | 0.27            |
> | **Math**         | AMPS Hard        | 0.46                      | 0.50            |
>
>
> **In Summary**:
> Our focus on reasoning tasks underscores their broad applicability, alignment with the core principles of TextGrad and FL, and relevance to real-world applications. By incorporating additional experiments on diverse tasks, such as coding and problem-solving, in the revised version, we have demonstrated the broader generalizability of FedTextGrad. We appreciate your thoughtful feedback, which has strengthened the clarity and scope of our work.
>
> Reference:
> - [1] Huang, Jie, and Kevin Chen-Chuan Chang. "Towards reasoning in large language models: A survey." arXiv preprint arXiv:2212.10403 (2022).
> - [2] White, Colin, et al. "Livebench: A challenging, contamination-free llm benchmark." arXiv preprint arXiv:2406.19314 (2024).

---

> ### Author Response · Authors · 2024-11-25
>
> **Q2: Comparison with Other Methods: Why not compare directly with existing federated learning methods for LLMs?**
>
> Thank you for your thoughtful comment. Below, we clarify the unique context of our work and the reasoning behind the absence of direct comparisons with traditional FL methods.
>
> **1. Clarification of the Study’s Scope**:
> Our study addresses real-world applications of **LLM APIs**, which inherently lack support for **numerical gradient computation or differential loss calculation**. This constraint, detailed in **Line 13–15** and reiterated in the introduction (**Line 53–57**), defines the boundaries of our research. These limitations necessitate innovative solutions; the FedTextGrad framework addresses this by leveraging textual feedback to enable collaborative optimization in scenarios where numerical gradients are unavailable.
>
> **2. Infeasibility of Direct Comparisons**:
> Traditional FL methods, including **FedAvg** and the referenced **FedBiOT**, rely on numerical gradients and loss functions for optimization. In contrast, LLM APIs function as black-box systems, where such information is not accessible. This key distinction makes direct comparisons methodologically infeasible. To address this gap, our proposed FedTextGrad framework introduces a novel paradigm for federated optimization based on textual feedback, specifically designed to overcome the constraints of black-box LLM API scenarios. As a result, FedTextGrad represents a pioneering approach that extends the applicability of FL to previously inaccessible domains.
>
> **3. Comprehensive Experimental Comparisons in the New Setting**:
> While direct comparisons with **traditional numerical FL methods** are infeasible, we conducted **comprehensive experiments** to evaluate FedTextGrad within its unique setting. Our experiments have been recognized by reviewers for **comprehensiveness** and **systematic analysis**:
>   - Reviewer **iWmc** noted:
>     > “The paper thoroughly investigates key FL hyperparameters… This systematic analysis provides practical insights into optimizing FedTextGrad for different settings.”
>   - Reviewer **X2Ad** highlighted:
>     > “Numerous insights on learning in a federated learning setting without numerical gradients.”
>   - Reviewer **3fjj** commented:
>     > “The experiments in the paper are comprehensive.”
>
> These acknowledgments underscore the depth and practical relevance of our experimental evaluation.
>
> Specifically, we identified the key and generally applicable module in FedTextGrad—textual update aggregation—and compared its performance across three aggregation methods: concatenation, summarization, and summarization guided by the uniform information density hypothesis. Detailed analysis and insights from these comparisons are provided in the paper **Section 4.2 and 4.3**, offering a comprehensive evaluation of our framework’s performance.
>
> Thank you for highlighting this point, as it helps us clarify and better articulate the **scope** and **significance** of our work.
>
> ---
>
> **Q3: Model Diversity in Studies: Why was Llama-3.1-8B exclusively used, without testing on other models?**
>
> Thank you for your insightful question regarding the exclusive use of LLaMA 3.1-8B for prompt optimization and ablation studies. Below, we clarify our rationale:
>
> **1. Conducted Evaluation Across Diverse Models**:
> Our experiments included a range of LLMs, as shown in **Figure 3**: Closed-source models: GPT-4. Open-source models: LLaMA 3.1-70B, 8B, Gemma, and Qwen2. Among these, LLaMA 3.1-8B was chosen for ablation studies due to its optimal balance between performance and computational efficiency.
>
> **2. Why Focus on LLaMA 3.1-8B?**
> - **Balanced Performance**: LLaMA 3.1-8B achieves results comparable to larger models while significantly outperforming smaller ones.
> - **Resource Efficiency**: Its computational requirements make it ideal for FL scenarios, where clients often face resource constraints.
> - **Cost-Effectiveness**: LLaMA 3.1-8B aligns with our computational budget, enabling detailed and reproducible studies without compromising on validity.
>
> **3. Encouraging Broader Validation**:
> Evaluating a wider range of models is computationally intensive and costly. To address this limitation, we encourage validation by the broader research community, which may have greater resources to explore additional architectures. This collaborative approach promotes fairness for resource-constrained researchers and aligns with environmentally sustainable practices.
>
> **In Summary**:
> LLaMA 3.1-8B was selected for ablation studies due to its balance of performance, efficiency, and cost, making it a practical and effective choice. We thank you for your valuable feedback, which has helped us clarify this decision and highlight opportunities for further exploration.

---

> ### Author Response · Authors · 2024-11-27
>
> Dear Reviewer hPgp,
>
> We sincerely thank you for your thoughtful review and valuable feedback. We have carefully addressed each of your questions and provided detailed responses in our rebuttal. We believe these clarifications effectively address the concerns you raised and further highlight the strength of our contributions.
>
> If our responses satisfactorily resolve your concerns, we would greatly appreciate it if you could consider updating your score.
>
> Please feel free to reach out if you have any further questions or need additional clarifications—we would be more than happy to discuss them in detail.
>
> Thank you once again for your time, effort, and thoughtful engagement with our work

---

> ### Author Response · Authors · 2024-12-02
>
> Dear Reviewer,
>
> We sincerely thank you for your valuable feedback and the opportunity to address your concerns. We would like to kindly remind you that December 2 is the final day for discussions, and the deadline is quickly approaching. If you feel that our responses have resolved the concerns you previously raised, we would greatly appreciate it if you could consider raising your score accordingly.
>
> If you have any remaining questions or need further clarification, we would be delighted to provide additional information before the deadline.
>
> Thank you again for your time and thoughtful review.
>
> Best regards,
> The Authors
> ICLR 2025 Submission #6860

---

> > ### Comment · Reviewer_hPgp · 2024-12-03
> >
> > Thanks for the response. It has addressed my concerns. I will increase my  score.

---

> > > ### Author Response · Authors · 2024-12-03
> > >
> > > Thank you for recognizing that we have addressed all your concerns effectively. We sincerely appreciate your thoughtful and constructive feedback!

---

### Official Review · Reviewer_3fjj · 2024-11-04

**Soundness:** 2
**Presentation:** 3
**Contribution:** 2
**Rating:** 5
**Confidence:** 5

**Summary:**

This paper investigates the integration of textual gradients into federated learning environments, focusing on improving the optimization of large language models with privacy preservation. The authors propose a novel approach, Federated Textual Gradient (FedTextGrad), which adapts the concept of textual gradients for federated settings.

**Strengths:**

This paper proposes a concept called FedTextGrad for optimizing large language models by integrating text gradients. This method utilizes text feedback for model optimization in a federated environment, extending the application of federated learning to areas where numerical gradients are impractical or infeasible. This paper identifies and addresses key challenges in joint text gradient aggregation, such as maintaining basic information in distributed updates and managing prompt sizes to accommodate LLM API constraints.

**Weaknesses:**

Although the experiments in the paper are comprehensive, they mainly focus on inference tasks, which may not fully demonstrate the broad applicability of FedTextGrad in various fields.
The discussion on the limitations of the FedTextGrad method is somewhat insufficient. Although this article briefly discusses challenges such as prompt length management and information retention, it does not delve into potential limitations or scalability issues that may arise when deployed in larger, more heterogeneous environments.
Some parts of the manuscript are dense and highly technical, which may make it difficult for readers to understand. Using complex descriptions may obscure key points and findings.

**Questions:**

1. The quality of text gradients generated across different clients may fluctuate. How can we ensure the consistency and effectiveness of these text gradients, especially in cases of uneven data distribution or varying data quality?
2. The paper tests the proposed method primarily on reasoning tasks. Can you discuss or provide insights on how FedTextGrad might perform across different domains, where data might be more sensitive or heterogeneous?
3. Consider providing more detailed descriptions for the implementation of FedTextGrad, especially how textual gradients are computed and aggregated.
4. Elaborate on the practical implications of FedTextGrad. How can this method be integrated into existing federated learning frameworks? What are the practical challenges of deploying FedTextGrad in operational environments, and how can they be overcome?

---

> ### Author Response · Authors · 2024-11-25
>
> **W1/Q2: Justification on Tasks Selected and Their Diversity.**
>
> We appreciate the opportunity to clarify the rationale for focusing on reasoning tasks in our experiments and to clarify that the tasks under reasoning are already diverse. We also address the broader applicability of FedTextGrad.
>
> **1. Wide Applicability of Reasoning Tasks**:
> **Reasoning in LLMs** involves tasks like logical deduction, commonsense reasoning, and multi-hop question answering, requiring models to **go beyond pattern recognition to manipulate and infer complex information** [1]. Evaluating LLMs on reasoning tasks provides a comprehensive assessment of their capabilities and helps identify areas for targeted improvements, making it a **cornerstone of advancing LLM**. In our study, we evaluated FedTextGrad using diverse reasoning benchmarks, including **BBH Object Counting**, **Multi-step Arithmetic**, and **GSM8K**, which **encompass complex and varied scenarios**. These tasks serve as representative examples of real-world applications where textual gradients are highly impactful.
>
> **2. Reasoning Tasks Are Core to TextGrad and Highly Relevant to FL Setting**:
> **Reasoning Tasks Align with TextGrad’s Paradigm**. Reasoning tasks are particularly well-suited for evaluating TextGrad because they rely on step-by-step thought processes and can be optimized through iterative textual feedback and self-refinement—core elements of the TextGrad paradigm. This alignment is highlighted in Section 1 (see **Line 59 to 63**) and validated by the findings in the original TextGrad paper. Reasoning is actively-studied and complicated problems in LLM research.
>
> **Reasoning as a Key Challenge in LLM-based TextGrad Research**. Reasoning tasks also represent a significant area of active research in LLM development, as they are both challenging and crucial for applications requiring complex decision-making and understanding. By focusing on reasoning tasks, we ensure that TextGrad is tested in scenarios that reflect real-world FL applications, where iterative optimization and feedback mechanisms are essential.
>
> **Justification of Task Selection from TextGrad.** While the original TextGrad study included tasks such as coding, chemistry, and radiotherapy treatment, these were not included here for specific reasons: coding tasks involve single-prompt optimization, which differs fundamentally from federated learning’s collaborative framework, and adapting them would require significant modifications outside our scope; chemistry and radiotherapy datasets are not clearly publicly accessible, and their integration into FL would require specialized adjustments for domain-specific constraints and task heterogeneity. Our focus on reasoning tasks provides a generalizable and accessible benchmark for evaluating TextGrad in FL scenarios.
>
> **3. Expanded Scope in Revised Version**:
> While reasoning tasks remain central to our study, we recognize the importance of exploring additional scenarios to demonstrate FedTextGrad’s versatility. In response to this feedback, we have added experiments on diverse tasks, diverse and challenging tasks from **LiveBench** [2] in **Appendix D**, in the revised version. These results showcase the broader applicability of FedTextGrad while reinforcing its effectiveness in reasoning-centric applications.
>
> | **Category**    | **Dataset**      | **Centralized TextGrad** | **FedTextGrad** |
> |------------------|------------------|---------------------------|------------------|
> | **Reasoning**    | Spatial          | 0.53                      | 0.40            |
> |                  | Web of Lies      | 0.37                      | 0.30            |
> |                  | Zebra Puzzle     | 0.33                      | 0.27            |
> | **Math**         | AMPS Hard        | 0.46                      | 0.50            |
>
>
> **In Summary**:
> Our focus on reasoning tasks underscores their broad applicability, alignment with the core principles of TextGrad and FL, and relevance to real-world applications. By incorporating additional experiments on diverse tasks, such as coding and problem-solving, in the revised version, we have demonstrated the broader generalizability of FedTextGrad. We appreciate your thoughtful feedback, which has strengthened the clarity and scope of our work.
>
> Reference:
> - [1] Huang, Jie, and Kevin Chen-Chuan Chang. "Towards reasoning in large language models: A survey." arXiv preprint arXiv:2212.10403 (2022).
> - [2] White, Colin, et al. "Livebench: A challenging, contamination-free llm benchmark." arXiv preprint arXiv:2406.19314 (2024).

---

> ### Author Response · Authors · 2024-11-25
>
> **W2: Discussion on challenges like scalability and more heterogeneous settings**
>
> We agree with the reviewer on the importance of addressing the challenges of scalability and heterogeneity. To address this comment, we first wish to highlight our efforts to cover these topics in our original submission and then justify the extent of their discussion based on the focus of our study.
>
> **Performance Evaluated in Heterogeneous Federated Settings.** We would like to highlight our discussion in the original submission to demonstrate our efforts in covering these topics. As detailed in **Section 3.3** of our submission,  we have analyzed FedTextGrad’s performance under client heterogeneity through diverse hyper-parameter ablation studies. These experiments provide valuable insights into FedTextGrad’s adaptability in heterogeneous federated settings, laying a strong foundation for further exploration in more complex and heterogeneous scenarios.
>
> **Discussed on Scalability and Heterogeneity Challenges.** Additionally, we have included the **Discussion Section 5** in the original manuscript to address challenges related to scalability and more heterogeneous environments. This includes a detailed discussion of potential limitations and the key directions required to extend FedTextGrad’s applicability to larger-scale and more diverse federated learning settings.
>
> **Primary Objective: Introducing and Validating FedTextGrad.** While scalability and extreme heterogeneity are critical considerations, the primary objective of our work is to introduce FedTextGrad as a novel paradigm for  FL leveraging textual gradients, and to rigorously validate its feasibility and effectiveness in foundational scenarios. To this end, we conducted extensive evaluations on key aspects of FL (e.g., local epochs, batch size, number of clients, heterogeneity levels, and aggregation methods), supported by ablation studies and in-depth analyses. We are pleased that several reviewers have acknowledged these efforts.  Furthermore, we explicitly outline potential future directions to inspire subsequent research on scalability and extreme heterogeneity (e.g., communication cost).
>
> **Enhanced Discussion in Revised Manuscript.** Our work establishes the viability of textual gradients in FL and lays a robust groundwork for addressing such challenges in follow-up studies, and focusing on wide topics of FL in this study would detract from its primary objective. To address your concerns, we have enhanced our discussion on heterogeneous environments in the updated manuscript in **Section 5 Line 462-469** of our revised version.
>
> **In Summary**:
> Our current study focuses on establishing and validating FedTextGrad as a pioneering framework for federated optimization with textual gradients. While scalability and heterogeneity challenges remain important, we have outlined these as key areas for future research and expanded the discussion to provide additional insights. Thank you for your valuable feedback, which has helped us refine our discussion and improve the manuscript.
>
> ---
>
> **W3: Dense and Technical Writing: Complex descriptions obscure key points, making the manuscript harder to understand**
>
> Thank you for your feedback regarding the readability of the manuscript. The paper already features a well-structured format with explicit subsections and clearly highlighted key points for ease of navigation—a strength recognized by **Reviewer hPgp**, who noted, “Clarity: The paper is clear and well-structured.” Nonetheless, we acknowledge the potential to further enhance its readability and accessibility. To address this, we have implemented the following measures:
>
> **1. Highlighted Key Information**:
> Key concepts and findings are now emphasized using bold text and summaries, particularly in **Sections 1, 2, and 5**. This makes it easier for readers to quickly locate and understand critical points.
>
> **2. Clear Takeaways in the Experimental Sections**:
> We have distilled the key results and insights from **Sections 3 and 4** into concise summaries. These refinements ensure that readers can readily grasp the main findings without being overwhelmed by technical details.
>
> These improvements enhance the manuscript’s clarity and accessibility, ensuring that key points and results are prominently and effectively communicated. We appreciate your valuable feedback, which has guided us in making these adjustments.

---

> ### Author Response · Authors · 2024-11-25
>
> **Q1: Consistency of Text Gradients: How to ensure consistent and effective gradients amid uneven data distribution and quality?**
>
> Thank you for raising this insightful question. As noted in **Section 3 (Lines 257–258)**, we employ a stability mechanism where, after each iteration, the same batch is evaluated in a loop, and the prompt is updated only if the performance does not drop compared to the previous non-updated version. This approach effectively stabilizes text gradient fluctuations among clients by ensuring that only beneficial updates are retained.
>
> Additionally, this mechanism has demonstrated effectiveness in heterogeneous settings of FedTextGrad, where varying data distributions and quality could otherwise amplify inconsistencies. By integrating this stabilization trick, we enhance the robustness of textual gradients across diverse client environments.
>
> We hope this clarifies our approach to addressing gradient consistency and its applicability in heterogeneous scenarios. Thank you again for your valuable question.
>
>
> ---
>
> **Q3: Implementation Details: Provide more clarity on how textual gradients are computed and aggregated.**
>
> Thank you for your suggestion to provide more detailed descriptions of the implementation of FedTextGrad. We would like to clarify that the implementation of the entire FedTextGrad procedure is thoroughly presented in the manuscript:
> **1. Overall Procedure**: The workflow of FedTextGrad is illustrated comprehensively in **Figure 1**, providing a clear overview of the process.
>
> **2. Textual Gradient Computation**: Detailed explanations are provided in **Section 2.1**, specifically in the paragraphs titled Backpropagation of TextGrad and LLMs-as-Optimizers in TextGrad, which outline the steps for generating textual gradients.
>
> **3. Textual Aggregation**: The aggregation process is described in **Section 2.3 Step 2 and Algorithm 1**, with an example prompt for summarization included in **Appendix C** to further clarify the procedure.
>
> These sections, including **Figure 1**, **Section 2.1**, **Section 2.3 Step 2**, **Algorithm 1**, and **Appendix C**, collectively offer a detailed and clear description of the textual gradient computation and aggregation implementation of FedTextGrad. We hope this addresses your concerns, and we are happy to incorporate additional clarifications if needed.
>
> ---
>
> **Q4: Practical Integration: How can FedTextGrad be integrated into existing frameworks, and what practical challenges arise?**
>
> Thank you for your thoughtful question. Below, we elaborate on the practical implications of FedTextGrad, its integration into existing FL frameworks, and the challenges of deploying it in real-world environments, along with potential solutions.
>
> **1. Practical Implications of FedTextGrad**:
> FedTextGrad is specifically designed for black-box LLM APIs in decentralized settings, as outlined in **Section 1 (Lines 55–57)**. It is particularly suitable for scenarios where numerical gradients are unavailable, enabling optimization through textual feedback. This makes FedTextGrad applicable to a variety of real-world use cases, including:
> - Collaborative Medical Research: Hospitals can collaboratively optimize LLM-based clinical decision support systems without sharing sensitive patient data. For example, hospitals with unique datasets can refine diagnostic prompts while preserving privacy, as demonstrated in our dataset experiments.
> - Personalized Education Tools: Federated learning systems in education can enhance reasoning and problem-solving prompts by leveraging unshared personal data to adapt to diverse learning patterns, as shown in our experimental results.
>
> **2. Integration with Existing FL Frameworks**:
> We would like to clarify that FedTextGrad is not intended as a module or extension to existing FL frameworks but rather as a **foundational and entirely novel framework**. It is specifically designed for settings where numerical gradients are inaccessible, loss functions are non-differentiable, and feedback is provided in textual form. This distinct paradigm addresses scenarios that traditional FL frameworks cannot accommodate. We have emphasized this unique positioning in the **Abstract (Lines 15–20)** and **Section 1 Introduction (Lines 72–77)** of the manuscript to clearly differentiate FedTextGrad from existing FL approaches.
>
> **3.Practical Challenges and Solutions**:
> We mentioned the practical challenges and possible directions in **Section 5** (we can summarize what we talked about). We also would like to highlight that using FedTextGrad on top of black-box LLM API make the use of FL easier and more interpretable in training and deployment as users can directly call APIs to launch training and inference, without needs of delving into complex neural network numerical training pipelines.

---

> ### Author Response · Authors · 2024-11-27
>
> Dear Reviewer 3fjj,
>
> We sincerely thank you for your thoughtful review and valuable feedback. We have carefully addressed each of your questions and provided detailed responses in our rebuttal. We believe these clarifications effectively address the concerns you raised and further highlight the strength of our contributions.
>
> If our responses satisfactorily resolve your concerns, we would greatly appreciate it if you could consider updating your score.
>
> Please feel free to reach out if you have any further questions or need additional clarifications—we would be more than happy to discuss them in detail.
>
> Thank you once again for your time, effort, and thoughtful engagement with our work

---

> ### Author Response · Authors · 2024-12-02
>
> Dear Reviewer,
>
> We sincerely thank you for your valuable feedback and the opportunity to address your concerns. We would like to kindly remind you that December 2 is the final day for discussions, and the deadline is quickly approaching. If you feel that our responses have resolved the concerns you previously raised, we would greatly appreciate it if you could consider raising your score accordingly.
>
> If you have any remaining questions or need further clarification, we would be delighted to provide additional information before the deadline.
>
> Thank you again for your time and thoughtful review.
>
> Best regards,
> The Authors
> ICLR 2025 Submission #6860

---

### Official Review · Reviewer_X2Ad · 2024-11-04

**Soundness:** 3
**Presentation:** 3
**Contribution:** 3
**Rating:** 6
**Confidence:** 4

**Summary:**

The authors present a novel approach to analyze the potential and pitfalls of integrating methods like TextGrad (Yuksekgonul et al., 2024) into a federated learning context. In this setup, each client initiates a prompt and refines it locally using its data and local LLM. The optimized local prompts are then uploaded to a server, which aggregates the prompts and redistributes them back to the clients. The main aggregation strategies considered are concatenation and summarization, with an additional focus on optimizing aggregation by prioritizing words with the highest information content.

The authors offer several insights:
- Accuracy vs. number of local steps performed before sending the final prompt to the server
- Accuracy vs. number of clients
- Accuracy vs. batch size
- Impact of using different LLMs, e.g., GPT-4 vs. LLaMA
- Cost implications of concatenation in terms of $
- Insights on summarization vs. concatenation

One potential area for improvement, although I understand is challenging due to the novelty of the topic, is a comparison with previous federated learning or prompt optimization solutions. For example:
- A comparison with traditional Federated Learning based on numerical gradients, such as Communication-Efficient Learning of Deep Networks from Decentralized Data (2023) or FedBiOT: LLM Local Fine-Tuning in Federated Learning without Full Model (2023). If these comparisons are not feasible, it would be helpful to clarify why.
- A direct comparison with the centralized version (TextGrad).

**Strengths:**

- Presentation of a novel framework for handling prompt-based learning in a federated learning context.
- Numerous insights on learning in a federated learning setting without numerical gradients.

**Weaknesses:**

- lack of comparisons w.r.t. previous works on federated learning

**Questions:**

Although the topic is new, authors should try to do the following things if possible:
- A comparison with traditional Federated Learning based on numerical gradients, such as Communication-Efficient Learning of Deep Networks from Decentralized Data (2023) or FedBiOT: LLM Local Fine-Tuning in Federated Learning without Full Model (2023). If these comparisons are not feasible, it would be helpful to clarify why.
- A direct comparison with the centralized version (TextGrad).

---

> ### Author Response · Authors · 2024-11-25
>
> **W1/Q1: Justification on the Selections of Comparisons**
>
> Thank you for your thoughtful comment. Below, we clarify the unique context of our work and the reasoning behind the absence of direct comparisons with traditional FL methods.
>
> **1. Clarification of the Study’s Scope**:
> Our study addresses real-world applications of **LLM APIs**, which inherently lack support for **numerical gradient computation or differential loss calculation**. This constraint, detailed in **Line 13–15** and reiterated in the introduction (**Line 53–57**), defines the boundaries of our research. These limitations necessitate innovative solutions; the FedTextGrad framework addresses this by leveraging textual feedback to enable collaborative optimization in scenarios where numerical gradients are unavailable.
>
> **2. Infeasibility of Direct Comparisons**:
> Traditional FL methods, including **FedAvg** and the referenced **FedBiOT**, rely on numerical gradients and loss functions for optimization. In contrast, LLM APIs function as black-box systems, where such information is not accessible. This key distinction makes direct comparisons methodologically infeasible. To address this gap, our proposed FedTextGrad framework introduces a novel paradigm for federated optimization based on textual feedback, specifically designed to overcome the constraints of black-box LLM API scenarios. As a result, FedTextGrad represents a pioneering approach that extends the applicability of FL to previously inaccessible domains.
>
> **3. Comprehensive Experimental Comparisons in the New Setting**:
> While direct comparisons with **traditional numerical FL methods** are infeasible, we conducted **comprehensive experiments** to evaluate FedTextGrad within its unique setting. Our experiments have been recognized by reviewers for **comprehensiveness** and **systematic analysis**:
>   - Reviewer **iWmc** noted:
>     > “The paper thoroughly investigates key FL hyperparameters… This systematic analysis provides practical insights into optimizing FedTextGrad for different settings.”
>   - Reviewer **3fjj** commented:
>     > “The experiments in the paper are comprehensive.”
>   - Reviewer **hPgp** highlighted:
>     > “The paper provides a thorough analysis of the proposed method’s feasibility and identifies key challenges.”
>
> These acknowledgments underscore the depth and practical relevance of our experimental evaluation.
>
> Specifically, we identified the key and generally applicable module in FedTextGrad—textual update aggregation—and compared its performance across three aggregation methods: concatenation, summarization, and summarization guided by the uniform information density hypothesis. Detailed analysis and insights from these comparisons are provided in the paper **Section 4.2 and 4.3**, offering a comprehensive evaluation of our framework’s performance.
>
> Thank you for highlighting this point, as it helps us clarify and better articulate the **scope** and **significance** of our work.
>
> ---
>
> **Q2: Comparison with Centralized TextGrad – Already Addressed in Our Submission**
>
> **Response:**
>
> Thank you for highlighting the importance of comparing **FedTextGrad** with its centralized counterpart, **TextGrad**. We would like to clarify that this comparison is already presented in **Figure 3** of the paper.
>
> **Comparison Details**:
> - **Figure 3a** shows the performance of **TextGrad** in a **centralized setting** across various tasks.
> - **Figure 3b** presents the results for **FedTextGrad** in **federated settings**.
>
> The side-by-side comparison highlights the **performance gap** between the two approaches. As expected, centralized TextGrad** achieves higher accuracy, as it is not subject to federated constraints like communication overhead and client heterogeneity.

---

> ### Author Response · Authors · 2024-11-27
>
> Dear Reviewer X2Ad,
>
> We sincerely thank you for your thoughtful review and valuable feedback. We have carefully addressed each of your questions and provided detailed responses in our rebuttal. We believe these clarifications effectively address the concerns you raised and further highlight the strength of our contributions.
>
> If our responses satisfactorily resolve your concerns, we would greatly appreciate it if you could consider updating your score.
>
> Please feel free to reach out if you have any further questions or need additional clarifications—we would be more than happy to discuss them in detail.
>
> Thank you once again for your time, effort, and thoughtful engagement with our work

---

> ### Author Response · Authors · 2024-12-02
>
> Dear Reviewer,
>
> We sincerely thank you for your valuable feedback and the opportunity to address your concerns. We would like to kindly remind you that December 2 is the final day for discussions, and the deadline is quickly approaching. If you feel that our responses have resolved the concerns you previously raised, we would greatly appreciate it if you could consider raising your score accordingly.
>
> If you have any remaining questions or need further clarification, we would be delighted to provide additional information before the deadline.
>
> Thank you again for your time and thoughtful review.
>
> Best regards,
> The Authors
> ICLR 2025 Submission #6860

---

### Official Review · Reviewer_iWmc · 2024-11-06

**Soundness:** 2
**Presentation:** 2
**Contribution:** 3
**Rating:** 6
**Confidence:** 3

**Summary:**

This paper introduces FedTextGrad, a novel paradigm designed to extend the capabilities of federated learning (FL) by incorporating textual gradients rather than traditional numerical ones. Building on TextGrad, an approach for LLM-based prompt optimization through text-based feedback, this work explores how such textual gradients can function within FL frameworks. The motivation behind this approach is to enable FL applications where numerical loss functions or gradients are impractical or unavailable, such as when working with black-box LLM APIs. FedTextGrad allows clients to upload optimized prompts—derived from textual feedback during local training—to a central server, which then aggregates these prompts and redistributes a global prompt to clients. This method adapts FL for decentralized and privacy-sensitive environments, expanding potential applications.

The paper identifies unique challenges in aggregating textual gradients across clients, particularly in retaining essential context within the aggregated prompts. Traditional methods like concatenation often lead to prompts that exceed LLM token limits, while summarization can compromise performance by overly compressing information. To address this, the authors propose an enhancement based on the Uniform Information Density (UID) principle, which balances information distribution within aggregated prompts to maintain a coherent context without excessive length. This UID-informed summarization improved global prompt quality.

Experiments are conducted to evaluate FedTextGrad across various reasoning tasks, including object counting and arithmetic problems, using different LLM architectures such as LLaMA and GPT-4. The results demonstrate that key FL hyperparameters, like local steps, batch size, and the number of clients, greatly impact FedTextGrad’s performance. Increasing local steps can enhance local adaptation but risks misalignment with the global prompt, while adding clients initially improves performance by introducing data diversity but eventually leads to synchronization challenges. These findings underscore the importance of carefully tuning FL parameters to balance local and global model alignment in a textual gradient setting.

While the study offers promising insights into applying textual gradients in FL, it acknowledges several limitations, particularly in terms of privacy and security. Textual gradients inherently carry more contextual information than numerical gradients, posing privacy risks when shared among clients and servers. Though the paper suggests differential privacy and encryption as potential solutions, it lacks an experimental exploration of these methods, leaving privacy protection as a critical area for future research.

**Strengths:**

1) The introduction of FedTextGrad as a framework to incorporate textual gradients into FL represents a novel approach, particularly valuable in settings where numerical gradients are unavailable, such as black-box LLM applications.
2) The paper addresses a core challenge in aggregating textual data—preserving essential context without exceeding token limits—by proposing a UID-based summarization approach. This innovative method helps maintain critical information balance across prompts, solving a major limitation in using textual gradients and improving FedTextGrad's scalability and robustness in handling prompt length constraints in federated settings.
3) The paper thoroughly investigates key FL hyperparameters, including local steps, batch size, and the number of clients, assessing their impact on performance. This systematic analysis provides practical insights into optimizing FedTextGrad for different settings, demonstrating the framework's adaptability and offering a foundation for further research and application in federated environments with textual gradients.

**Weaknesses:**

1) The paper identifies privacy risks with textual gradients but does not provide or test concrete methods to protect sensitive information, which is crucial for FL applications in privacy-sensitive domains.
2) The experiments are primarily conducted on a few large LLMs, restricting insights into how FedTextGrad performs across diverse architectures, particularly smaller models in resource-constrained settings.
3) The UID-based summarization method helps with prompt aggregation but has limitations in fully retaining information, especially with high client heterogeneity and complexity, potentially affecting scalability. The effectiveness of UID summarization under varying levels of client heterogeneity and task complexity is not thoroughly explored.

**Questions:**

1) Has dynamic switching between aggregation methods, based on task complexity, been considered?
2) How effectively does UID summarization capture essential information across tasks?

---

> ### Author Response · Authors · 2024-11-25
>
> **W1: Justification of our focus and the discussion on privacy**
>
> We appreciate the opportunity to address your question. First, we would like to clarify our primary focus and contribution. Additionally, we add more detailed discussion in the future directions section regarding the limitations of applying existing privacy-preserving methods to this new paradigm.
>
> **1. The Focus of Our Study**:
> Our work explores the **utility and challenges** of incorporating **textual gradients** into **FL**, establishing a foundational framework for this novel approach (**Line 19**). While privacy is a critical consideration, the primary objective of this study is to assess the feasibility, methodology, and performance of FedTextGrad. As such, the implementation or testing of privacy-preserving methods falls outside the scope of this foundational work.
>
> **2. Identified Privacy Challenges and Future Directions**:
> We **explicitly acknowledged the challenges** of applying traditional privacy-preserving techniques to textual gradients in Section 5 (**Line 501–511**). The complexity and context-rich nature of textual gradients introduce unique risks that existing approaches, such as differential privacy or secure multi-party computation, are not fully equipped to address. Inspired by precedents like DP-FedAvg [1] following FedAvg [2], we view the development of privacy-preserving adaptations of FedTextGrad as a significant direction for future research. To enhance this discussion, we have expanded **Appendix A.1 (Paragraph 3)** to include a more detailed survey of privacy-preserving methods for LLM prompting and provide more actionable and relevant guidance for integrating privacy into FedTextGrad.
>
> **In Summary**:
> While privacy challenges remain a critical area, our study establishes a foundational framework for exploring textual gradients in FL. Specifically, it demonstrates the feasibility, methodology, and performance of FedTextGrad in addressing black-box optimization scenarios where traditional numerical gradient methods are inapplicable. Privacy is considered a key direction for future research. The revised manuscript includes an expanded discussion to guide future efforts in addressing these challenges. Thank you for your valuable feedback, which has helped us strengthen this aspect of our work.
>
> References:
> - [1] Noble, M., Bellet, A., & Dieuleveut, A. (2022, May). Differentially private federated learning on heterogeneous data. In the International Conference on Artificial Intelligence and Statistics (pp. 10110-10145). PMLR.
> - [2] McMahan, B., Moore, E., Ramage, D., Hampson, S., & y Arcas, B. A. (2017, April). Communication-efficient learning of deep networks from decentralized data. In Artificial intelligence and statistics (pp. 1273-1282). PMLR.

---

> ### Author Response · Authors · 2024-11-25
>
> **W2: Evaluation of diverse LLMs, especially smaller models.**
>
> Thank you for your valuable suggestions regarding the evaluation of FedTextGrad across various architectures, particularly smaller models in resource-constrained environments. In this context, we would like to highlight our experiments, which provide new insights on how to effectively accommodate resource-constrained settings.
>
> **1. Clarification: Evaluated with Diverse and Smaller Models**
> We have already included experiments with **diverse and smaller LLMs**, such as LLaMA 3.1-8B, Gemma-2-9B, and Qwen-2-7B, as shown in **Figure 3** of the original paper. These experiments revealed a notable performance drop in smaller, open-source LLMs compared to their larger counterparts, indicating that current larger models are better suited for TextGrad, while much smaller models are less compatible with the framework. This observation aligns with prior findings in TextGrad, which demonstrate that TextGrad performance improves with model scale.
>
> **2. Transferability to Smaller Models Deployment**
> To address the potential for deployment in **resource-constrained** settings, we explored a promising method: **transferring prompts optimized on larger LLMs to smaller LLMs for deployment**. Additional experiments are detailed in **Appendix E** of the revised manuscript. The results of the Object Counting and Multi-step Arithmetic tasks demonstrate that prompts fine-tuned on larger LLMs can be effectively reused by smaller models, achieving significantly better performance than initial prompts without requiring further optimization. This finding has substantial practical implications, as it enables the use of larger LLMs for training, leveraging their optimization capabilities, while deploying smaller, resource-efficient models for inference in real-world applications.
>
> Again, we appreciate your feedback that highlights the importance of expanding FedTextGrad’s adaptability to a wider range of model architectures, including smaller LLMs. In response, we have included this as a key future research direction in the discussion section (see **Appendix A.2** of the revised paper). Specifically, we outline strategies for deploying smaller models in resource-constrained settings.
>
> | **Task**              | **Initial Prompt ↑** | **Transferred Prompt ↑** | **Performance Change** |
> |-----------------------|-----------------------|---------------------------|-------------------------|
> | Object Counting       | 0.66                 | **0.69**                  | +0.03                  |
> | Multi-step Arithmetic | 0.51                 | **0.66**                  | +0.15                  |
> | GSM8K                 | **0.80**             | 0.72                      | -0.08                  |
>
>
> **In Summary**:
> We did include the smaller LLM performance evaluation. To further resolve the issue of deployment of smaller LLM, we have now conducted additional experiments to evaluate FedTextGrad’s applicability to smaller models through prompt transferability, demonstrating its potential for resource-constrained deployments. Furthermore, we have expanded the discussion of future directions to emphasize the importance of extending this framework to a broader range of architectures.

---

> ### Author Response · Authors · 2024-11-25
>
> **W3: UID summarization’s effectiveness under diverse heterogeneities and complexities is underexplored.**
>
> Thank you for raising concerns about the limitations of UID-based summarization in handling high client heterogeneity and task complexity. We would like to provide clarification and additional evidence to address these points:
>
> **1. Clarification: Effectiveness Across Diverse Tasks**
> The effectiveness of UID summarization is demonstrated across diverse tasks, as shown in **Figure 6(b)**. These results highlight its ability to retain critical information while maintaining performance across varying task complexities.
>
> **2. Experiment: UID Summarization Evaluation Under Client Heterogeneity**
> To specifically address your concern regarding client heterogeneity, we conducted additional experiments to evaluate UID summarization under data heterogeneity. The results, presented in **Appendix F**, show that UID-based summarization performs effectively under task level of client heterogeneity.
>
> | **Method**          | **B = 1**         | **B = 3**         | **B = 10**        |
> |----------------------|-------------------|-------------------|-------------------|
> | Summarization        | 0.73 (0.03)      | 0.78 (0.02)      | 0.72 (0.03)      |
> | UID Summarization    | **0.75 (0.02)**  | **0.79 (0.02)**  | **0.74 (0.03)**  |
>
>
> ---
>
> **Q1: Has dynamic aggregation switching based on task complexity been considered?**
>
> Thank you for raising this insightful question. While we did not initially focus on dynamic aggregation switching, we recognize its potential as a promising extension. Implementing such strategies involves challenges, including defining clear switching criteria and accurately pre-quantifying task complexity in federated learning settings.
>
> To address this, we conducted an experiment to evaluate the feasibility of dynamic switching between concatenation and summarization during the federated aggregation stage. Specifically, when concatenated prompts exceed a pre-defined context window, summarization is applied. The results are detailed in **Appendix G** in our revision. The results demonstrate that the proposed dynamic aggregation method outperforms summarization in certain scenarios but underperforms summarization in others. However, it does not surpass concatenation when the window length is feasible. We found that establishing appropriate switching criteria is critical, presenting an intriguing avenue for future research.
>
> We sincerely appreciate your insightful comments and have highlighted potential future exploration directions in **Appendix G** of our revised manuscript.
>
> ---
>
> **Q2: How effectively does UID summarization retain essential information?**
>
> Thanks for the question. Compared with summarization, UID summarization is specifically designed to enhance uniformity in the distribution of information across prompts. While summarization methods inherently involve some information reduction compared to concatenation (as shown in **Figure 5**), UID summarization does not introduce any additional information loss beyond standard summarization methods by design.
>
> As described in **Line 91-92** of our original submission, we pointed out that UID aims “to ensure more balanced information distribution across the summarized global prompt.” As a result, it “preserves key information from each client while preventing overcompression” (see **Line 446-447** of our original submission).
>
> Firstly, the effectiveness of UID summarization in retaining critical information is demonstrated through task performance improvements over standard summarization. As detailed in **Section 4** and our new experiments presented under **W3**, UID summarization achieves higher task performance, indicating that it preserves essential information effectively. Additionally, we measured the mean surprisal value (represents the average information density) of output text generated from standard summarization versus UID summarization. This analysis, included in **Appendix H**, shows that UID summarization yields higher mean surprisal values, further highlighting UID summarization’s ability to retain critical information while ensuring better uniformity.

---

> ### Author Response · Authors · 2024-11-27
>
> Dear Reviewer iWmc,
>
> We sincerely thank you for your thoughtful review and valuable feedback. We have carefully addressed each of your questions and provided detailed responses in our rebuttal. We believe these clarifications effectively address the concerns you raised and further highlight the strength of our contributions.
>
> If our responses satisfactorily resolve your concerns, we would greatly appreciate it if you could consider updating your score.
>
> Please feel free to reach out if you have any further questions or need additional clarifications—we would be more than happy to discuss them in detail.
>
> Thank you once again for your time, effort, and thoughtful engagement with our work

---

> ### Author Response · Authors · 2024-12-02
>
> Dear Reviewer,
>
> We sincerely thank you for your valuable feedback and the opportunity to address your concerns. We would like to kindly remind you that December 2 is the final day for discussions, and the deadline is quickly approaching. If you feel that our responses have resolved the concerns you previously raised, we would greatly appreciate it if you could consider raising your score accordingly.
>
> If you have any remaining questions or need further clarification, we would be delighted to provide additional information before the deadline.
>
> Thank you again for your time and thoughtful review.
>
> Best regards,
> The Authors
> ICLR 2025 Submission #6860

---

### Meta-Review · Area_Chair_mYd1 · 2024-12-21

**Metareview:**

The authors propose a novel framework for exploring the potential and challenges of incorporating methods like TextGrad (Yuksekgonul et al., 2024) within a federated learning context. In this framework, each client generates an initial prompt and refines it locally using its own data and local LLM. The optimized prompts are then transmitted to a central server, where they are aggregated and redistributed to the clients. The study evaluates two primary aggregation strategies: concatenation and summarization, with a particular emphasis on optimizing aggregation by prioritizing words with the highest information content.

+ The studied problem is important.
+ The proposed methods are well presented and explained

- The experiments can be further improved

**Additional Comments On Reviewer Discussion:**

In the rebuttal period, the authors have provided detailed responses. I have also carefully read the comments of the reviewers. I find that reviewers iWmc and X2Ad present many significant strengths of the paper, while the negative points can be easily addressed in the final version. I think this paper can be accepted.

---

### Decision · Program_Chairs · 2025-01-22

Accept (Poster)